# Sulfur defect engineering controls Li$_2$S crystal orientation towards dendrite-free lithium metal batteries

Jin-Xia Lin [1,9], Peng Dai[1,9], Sheng-Nan Hu[1,9], Shiyuan Zhou [1], Gyeong-Su Park [2,3], Chen-Guang Shi[1], Jun-Fei Shen[1], Yu-Xiang Xie[1], Wei-Chen Zheng[1], Hui Chen[1], Shi-Shi Liu[1], Hua-Yu Huang[1], Ying Zhong[1], Jun-Tao Li[4], Rena Oh [5] ✉, Xiaoyang Jerry Huang[6], Wen-Feng Lin [7] ✉, Ling Huang [1,8] ✉ & Shi-Gang Sun [1,6,8] ✉

Controlling nucleation and growth of Li is crucial to avoid dendrite formation for practical applications of lithium metal batteries. Li$_2$S has been exemplified to promote Li transport, but its crystal orientation significantly influences the Li deposition behaviors. Here, we investigate the interactions between Li and various surface structures of Li$_2$S, and reveal that the Li$_2$S(111) plane exhibits the highest Li affinity and the lowest diffusion barrier, leading to dense Li deposition. Using sulfur defect engineering for Li$_2$S crystal orientation control, we construct three-dimensional vertically oriented Li$_2$S(111)@Cu nanorod arrays as a Li metal electrode substrate and identify a substrate-dependent Li nucleation process and a facet-dependent growth mode. Furthermore, we demonstrate the versatility of the Li$_2$S(111)@Cu substrate when paired with two positive electrodes: achieving an initial discharge capacity of 138.8 mAh g$^{-1}$ with 88% capacity retention after 400 cycles at 83.5 mA g$^{-1}$ with LiFePO$_4$, and an initial discharge capacity of 181 mAh g$^{-1}$ with 80% capacity retention after 160 cycles at 60 mA g$^{-1}$ with commercial LiNi$_{0.8}$Co$_{0.1}$Mn$_{0.1}$O$_2$ positive electrode (4 mAh cm$^{-2}$).

Lithium (Li) metal batteries (LMBs) have been attracting increasing studies because of their impressively high theoretical specific capacity up to 3860 mAh g$^{-1}$ and the most negative redox potential for the Li electrode at −3.04 V vs standard hydrogen electrode[1–3]. However, Li metal typically undergoes an uneven deposition/dissolution process, leading to dendritic growth and resulting in a thick and rough solid electrolyte interphase (SEI)[4,5]. As a result, these Li dendrites cause low Coulombic efficiency (CE), rapid capacity decay and severe safety issues, which seriously hinder the practical applications of LMBs[6,7].

To date, Li$_2$S is particularly noteworthy due to its high Li ion conductivity[8–10]. More importantly, for these studies with various substrate structure designs, an excellent Li$_2$S artificial SEI has been primarily credited to be the main reasons for the improved

[1]State Key Laboratory of Physical Chemistry of Solid Surfaces, Collaborative Innovation Center of Chemistry for Energy Materials, College of Chemistry and Chemical Engineering, Xiamen University, Xiamen, China. [2]Institute of Next-Generation Semiconductor Convergence Technology, Daegu Gyeongbuk Institute of Science and Technology (DGIST), Daegu, Republic of Korea. [3]Department of Materials Science and Engineering and Research Institute of Advanced Materials, Seoul National University, Seoul, Republic of Korea. [4]College of Energy, Xiamen University, Xiamen, China. [5]School of Energy and Power Engineering, Chongqing University, Chongqing, China. [6]Center of Advanced Electrochemical Energy, Institute of Advanced Interdisciplinary Studies, School of Chemistry and Chemical Engineering, Chongqing University, Chongqing, China. [7]Department of Chemical Engineering, Loughborough University, Loughborough, Leicestershire, UK. [8]Innovation Research Institute in Advanced Electronic Chemicals of Quzhou, Zhejiang, China. [9]These authors contributed equally: Jin-Xia Lin, Peng Dai, Sheng-Nan Hu. ✉e-mail: rena_oh@cqu.edu.cn; w.lin@lboro.ac.uk; huangl@xmu.edu.cn; sgsun@xmu.edu.cn

performance of LMBs. In fact, most studies did not consider the substrate-dependent behavior of Li deposition.

In this work, we investigate the Li deposition behavior on different crystal facets of $Li_2S$. Our findings indicate that the close-packed $Li_2S(111)$ served as the most favorable facet for Li deposition attributed to its effective adsorption and rapid transport of Li. By employing defect engineering and in situ electrochemical lithiation of $Cu_2S$ with sulfur defects (denoted as $Cu_2S_x$), we prepared three-dimensional (3D) vertically aligned $Li_2S(111)@Cu$ nanorod arrays. These arrays served as a mixed ion/electron conductor, effectively balanced both electron and ion transport at the Li/substrate interface. Interestingly, we observed a transition in the Li nucleation and growth process, shifting from a progressive to an instantaneous mode when the substrate was changed from Cu to $Li_2S(111)@Cu$. The progressive pattern refers to the continuous generation of Li nuclei on the Cu substrate, couples with the growth of the pre-formed nuclei. This process often leads to the formation of non-uniform nucleation sites and uneven Li deposition, ultimately resulting in dendrite growth[11]. In contrast, in our work, by applying a $Li_2S(111)@Cu$ substrate, Li nuclei were instantaneously formed and evenly distributed due to the homogeneous surface conditions. The Coulombic efficiency (CE) of Li||Cu half-cell based on $Li_2S(111)@Cu$ substrate can reach up to 99.2%, displaying a stability over 500 cycles, which is five-fold and two-fold higher than that of Cu and $Li_2S@Cu$ substrates, respectively. Furthermore, the assembly of the $Li_2S(111)@Cu$ substrate into a full cell with $LiFePO_4$ positive electrode exhibited a good cycle stability by showcasing 400 cycles at 0.5 C with 88% capacity retention. Additionally, when combined with a commercial $LiNi_{0.8}Co_{0.1}Mn_{0.1}O_2$ positive electrode (4 mAh cm$^{-2}$), the full cell based on the $Li_2S(111)@Cu$ substrate retained 80% of its capacity at 0.3 C after 160 cycles, which is three-fold and two-fold improvements compared to the Cu and $Li_2S@Cu$ substrates, respectively.

## Results

### Interactions of Li with $Li_2S$ surfaces of different crystal orientations

$Li_2S$ is a widely studied inorganic material known for enabling fast Li ion transport in advanced lithium metal batteries (LMBs). However, the facet-dependent behavior of Li deposition on $Li_2S$ for stable batteries has not yet been conclusively demonstrated. To address this, we constructed representative facets of cubic $Li_2S$ with predominantly exposed (111), (110), and (311) planes, each featuring distinct close-packed arrangements, for our investigation (Supplementary Fig. 1 and Supplementary Data 1–3). Then the density functional theory (DFT) calculations were performed to understand the Li atom adsorption behaviors on these three representative facets. It was found that Li adsorption ability could significantly affect subsequent nucleation. On the $Li_2S(311)$ surface, two types of candidate sites for Li adsorption were examined, which showed corresponding binding energies ($E_{binding}$) of -0.63 eV and −1.42 eV, respectively (see Fig. 1a and Supplementary Table 1). In contrast, the $Li_2S(110)$ plane exhibited a relatively weaker interaction with Li than that of $Li_2S(311)$, with the highest $E_{binding}$ being −0.58 eV (see Fig. 1b, Supplementary Fig. 2 and Table 2). The weaker interaction on the $Li_2S(110)$ surface might be due to the ion arrangement appearing sparser, even though both the $Li_2S(311)$ and $Li_2S(110)$ planes have similar open and stepped site arrangements. By contrast, the $Li_2S(111)$ plane displays a tightly-packed hexagonal configuration, demonstrating the highest affinity among the three facets studied. The $E_{binding}$ recorded at spot C as shown in Fig. 1c is −5.37 eV, nearly 4 and 10 times higher than that of the $Li_2S(311)$ and $Li_2S(110)$ surfaces, respectively. It's worth noting that, besides spot C, both sites A and B on $Li_2S(111)$ plane showed a high affinity to Li with a large $E_{binding}$ of −4.23 eV and −5.14 eV, respectively (see Fig. 1c, Supplementary Fig. 3 and Table 3).

To further understand the underlying reasons for the different Li adsorption abilities on $Li_2S$ facets, Bader analysis was performed based on the Density Derived Electrostatic and Chemical 6 (DDEC6) approach for all potential sites on these three facets[12]. The results, as shown in Supplementary Tables 1–3 and Supplementary Data 4–6, reveal the charge transfer from Li to $Li_2S(111)$ surface as having the highest population for all the three facets studied. For example, 0.83 e$^-$ charge transfer took place from Li to $Li_2S(111)$ surface under its most stable configuration (spot C). This is higher than the charge transfer of 0.61 e$^-$ and 0.28 e$^-$ observed on the $Li_2S(311)$ and $Li_2S(110)$ surfaces, respectively. These results agree well with the $E_{binding}$ calculations, which show that the $Li_2S(111)$ plane displays the highest Li affinity. The corresponding charge density differences are provided in Fig. 1d–f to visualize the electronic transfer. They clearly show that electrons accumulate with high dispersion and frequency (depicted in yellow) on $Li_2S(111)$ surface, thus favoring Li adsorption and nucleation initiation. All these results suggest that the crystal orientation of the $Li_2S$ substrate impacted its ability to capture Li, with the order of favorable planes for lithophilic nucleation sites being (111) > (311) > (110).

Typically, the diffusion of Li over the adsorption sites is crucial during the Li deposition. Here, the climbing-image nudged elastic band (CI-NEB) method was further employed to calculate the energy barriers for Li diffusion on the three facets of $Li_2S$[13]. The diffusion pathways were determined based on the hopping of Li from the most stable state to another (Supplementary Note 1). The initial state (IS) and final state (FS) of diffusion path were chosen according to the energy minima, which are B1 to B2 for $Li_2S(311)$, C1 to C2 for $Li_2S(110)$, and $Li_2S(111)$, respectively. The calculation results are depicted in Fig. 1g–i, it was found that the $Li_2S(311)$ plane has an energy barrier ($E_{barrier}$) of 1.23 eV for Li transport, while the $Li_2S(110)$ and $Li_2S(111)$ surfaces have only trivial diffusion barriers of 0.23 and 0.18 eV, respectively. These values suggest that the crystal facets of the $Li_2S$ substrate influenced greatly on the Li transport, with a diffusion kinetics order of (111) > (110) >> (311). In summary, the DFT calculations demonstrate the crystal orientation-dependent nature of Li nucleation and growth on $Li_2S$ surfaces. The close-packed (111) plane not only promotes robust Li adsorption but also facilitates fast Li diffusion, which, in turn, favors a dense Li deposition.

### Synthesis and characterization of $Li_2S(111)@Cu$ substrate

Following up the discovery of the advantages of $Li_2S(111)$ for Li deposition, compared to the other two less compacted planes, we proceeded to construct $Li_2S(111)@Cu$ nanorods (NRs) on the surface of commercial Cu foam to serve as a designed Li negative electrode substrate, as schematically depicted in Fig. 2a, through architectural and sulfur (S) defect engineering[14]. Briefly, we grew vertically aligned $Cu_2S$ NRs on Cu foam (referred to as $Cu_2S@Cu$ foam) by using an alkaline corrosion and sulfidation method. After drying in vacuum, the as-prepared $Cu_2S@Cu$ NRs was annealed in argon (Ar) atmosphere at 280 °C for 60 min to obtain the S-deficient $Cu_2S@Cu$ NRs (denoted as $Cu_2S_x@Cu$ NRs). The introduction of defects leads to a properly tuned electronic structure, which is important for the subsequent in situ Li activation process in $Cu_2S_x$ system (see more in the Supplementary Note 2)[15]. The X-ray diffraction (XRD) pattern as shown in Supplementary Fig. 4a confirms the successful synthesis of $Cu_2S$ NRs and $Cu_2S_x$ on Cu foam, with characteristic reflections of chalcocite $Cu_2S$ (JCPDS No. 26-1116) detected in both samples. Moreover, a peak position shift can be seen in the enlarged pattern, attributed to the influence of S defects (Supplementary Fig. 4b and Supplementary Note 3)[16]. To qualitatively determine the existence of S defects beyond any artifacts, electron paramagnetic resonance (EPR) measurement was further conducted. In Fig. 2b, a strong EPR signal associated with S defects at $g = 2.003$ was observed at $Cu_2S_x$ NRs, confirming the generation of S defects during the annealing process[17]. To investigate the

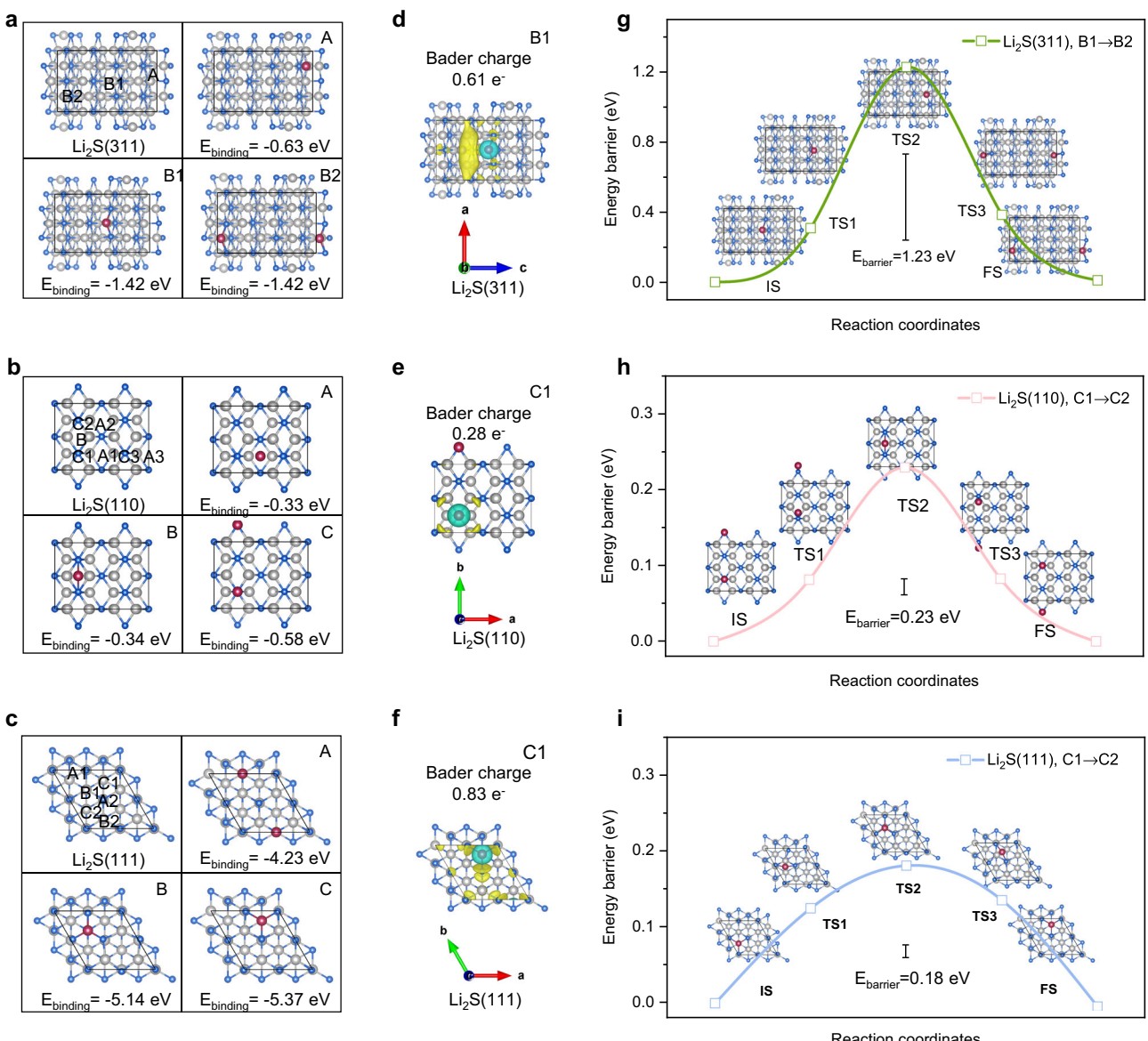

**Fig. 1 | DFT calculations of Li interactions with different facets of Li₂S. a–c** Binding energies. **d–f** Charge density difference plots. **g–i** Diffusion paths of Li on Li₂S(311) (**a**, **d**, **g**), Li₂S(110) (**b**, **e**, **h**) and Li₂S(111) (**c**, **f**, **i**) facets. Gray, blue, and pink represent Li, S and adsorbed Li atoms, respectively.

binding environment of the material, X-ray photoelectron spectroscopy (XPS) was also employed, and lower binding energies of Cu and higher binding energies of S were obtained at $Cu_2S_x@Cu$ compared with those at $Cu_2S@Cu$ (Supplementary Fig. 5), suggesting an increased electron density surrounding Cu atoms due to the decreased coordination of S[18]. These results align with our earlier characterizations that showed the existence of S defects. The atomic ratios of Cu:S for $Cu_2S$ and $Cu_2S_x$ from the XPS survey spectra were calculated to be 2.25:1 and 3.83:1, respectively (Supplementary Fig. 6), indicating a Cu-rich surface for $Cu_2S_x$. According to the percentage ratio of the removed S atoms to ideal number of S atoms, the amount of S defects in $Cu_2S_x$ was calculated to be 12.7 atomic% from the XPS data. To further analyze the chemical states and coordination environment between Cu and S, the X-ray absorption near-edge structure (XANES) spectra as shown in Supplementary Fig. 7a reveal a slight gap between $Cu_2S_x$ and $Cu_2S$ in their near-edge features, indicating a lower valence state of Cu in $Cu_2S_x$ than in $Cu_2S$. Fourier transform (FT) $k^3$-weighted extended X-ray absorption fine structure (EXAFS) spectra as shown in Supplementary Fig. 7b confirm the decreased coordination numbers (CN) of Cu-S in $Cu_2S_x$ compared with $Cu_2S$, with Cu-S distance of ~1.9 Å

in their outer shell scattering. All these results were visualized using wavelet transform (WT) analysis with simultaneous $k$ and $R$ space resolution. Furthermore, the quantitative CN was determined by fitting the Cu K-edge EXAFS curves at $k$ and $R$ space. As shown in Supplementary Fig. 8 and Table 4, the CN of Cu-S in $Cu_2S$ and $Cu_2S_x$ was found to be $4.3 \pm 0.3$ and $3.9 \pm 0.3$, respectively. Collectively, by applying multiple spectroscopic techniques, including XRD, EPR, XPS, and XAS, all results evidenced the formation of S defects in $Cu_2S_x$ and the change in the chemical environment.

Apart from the spectroscopy, microscopic characterizations using scanning electron microscopy (SEM) and transmission electron microscopy (TEM) were carried out to observe the morphology of synthesized materials. Supplementary Fig. 9 exhibits the pristine flat surface of bare Cu foam. The surface became rough after distributing the $Cu_2S$ NRs uniformly (see Supplementary Fig. 10a–d). TEM images in Supplementary Fig. 10e, f reveal that the structure of these nanorods, consisting of stacked $Cu_2S$ particles with an average diameter about 350 nm. High angle annular dark field scanning TEM (HAADF-STEM) with elemental mapping of $Cu_2S$ NRs was presented in Supplementary Fig. 10g, illustrating a homogeneous distribution of Cu and

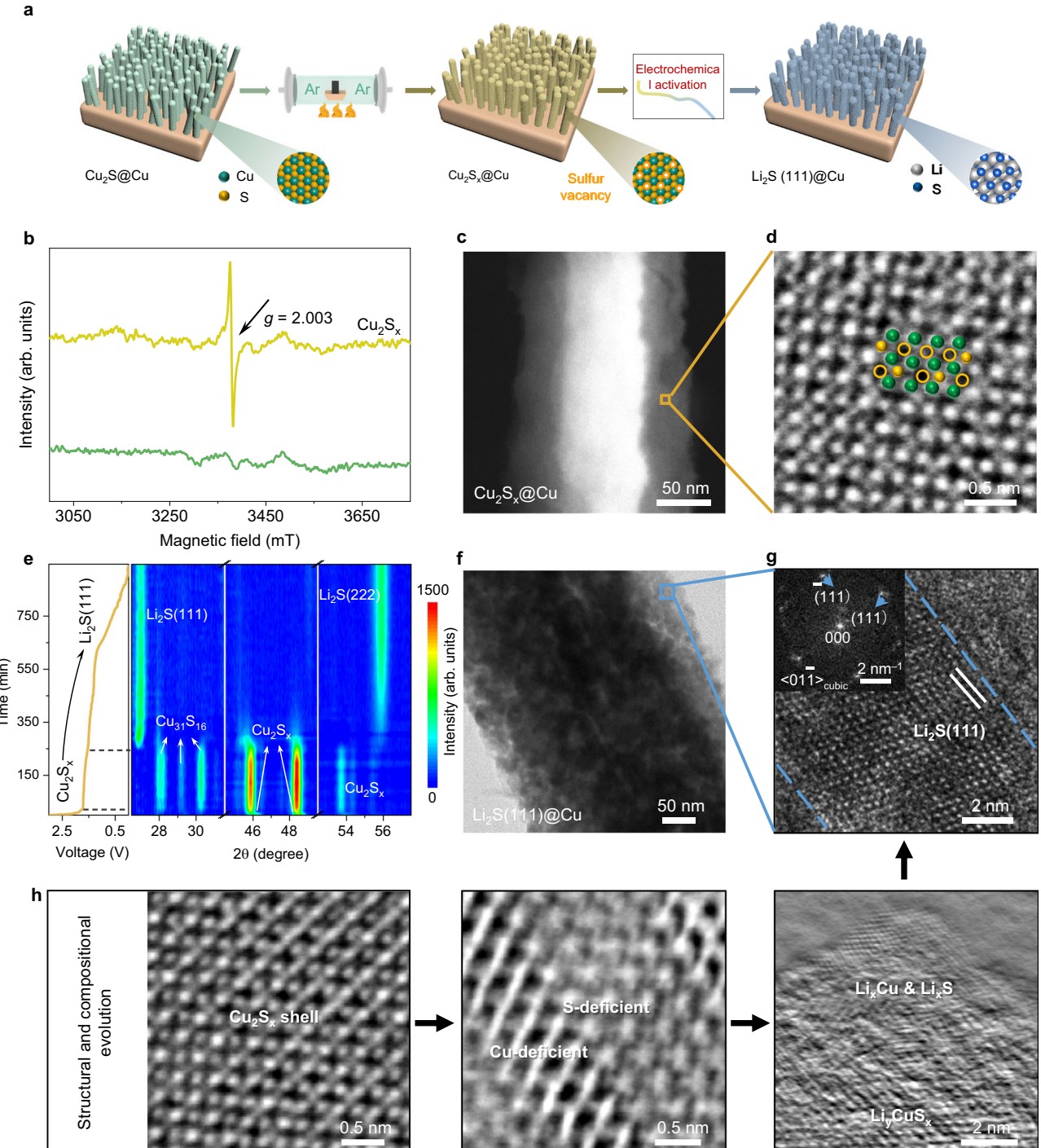

**Fig. 2 | Preparation and characterizations. a** Schematic representation for the design of $Li_2S(111)@Cu$ substrate. **b** EPR spectra of $Cu_2S_x$ and $Cu_2S$. **c** STEM image of $Cu_2S_x$ NR. **d** High-resolution iDPC-STEM image. **e** In situ XRD contour maps and corresponding discharge curve of forming $Li_2S(111)@Cu$ (Applied current density: 0.5 mA cm$^{-2}$, potential cutoff: 0.01 V, temperature: 25 ±1 °C). **f** TEM image of $Li_2S(111)$. **g** The corresponding HRTEM image, and the inset is the FFT diffractogram. **h** The structural and compositional evolution occurring during the lithiation.

S. High-resolution TEM (HRTEM), along with the corresponding fast Fourier transform (FFT) in Supplementary Fig. 10h, i, confirms the good crystallinity of $Cu_2S$ NRs. During the annealing process, the morphology of $Cu_2S@Cu$ NRs underwent slight changes, including a decreased diameter of $Cu_2S_x$ NRs (as seen in Supplementary Fig. 11). This diameter change leads to a reduced density of the $Cu_2S_x$ NRs, which could enrich the available space for Li deposition. In addition, the surface of $Cu_2S_x$ NRs became rougher and thus possibly resulting in an increased number of active sites during lithiation process.

Furthermore, we investigated microstructures of the $Cu_2S_x$ NRs using aberration-corrected TEM/STEM techniques. Our observations revealed that the region of the nanorod ~30 nm from the surface exhibited less atomic density, resulting in a darker image contrast in TEM images possibly due to defective structures additionally confirmed by HRTEM images with FFT diffractograms (see Supplementary Fig. 12a, b). The defective structures were also highlighted in Energy-dispersive X-ray spectroscopy (EDS) elemental maps showing a lower image contrast in the surface region (see Supplementary Fig. 12c).

Based on the EDS data, we roughly estimated the atomic ratio of Cu:S of $Cu_2S_x$, obtaining the S defect content of 14.5 atomic%, displayed in Supplementary Fig. 13, where the result followed the same trend as the previous XPS characterization (see Supplementary Fig. 6). To delve deeper into the atomic configuration of these defective structures, integrated differential phase contrast-STEM (iDPC-STEM) technique was used on the surface region marked in Fig. 2c. Notably, this method enabled the direct observation of S atoms and S defects at the atomic scale, with lower image contrast than Cu atoms, as shown in Fig. 2d. In contrast, HAADF-STEM was unable to visualize these S atoms and defects, as demonstrated in Supplementary Fig. 12d. These spectroscopic and microscopic results both demonstrate the successful preparation of $Cu_2S_x$@Cu NRs with abundant S defects. With this material in hand, we then proceeded the in situ Li$_2$S(111)@Cu substrate construction through electrochemical lithiation of $Cu_2S_x$@Cu NRs.

Galvanostatic discharge/charge (GDC) profiles were initially employed to detect the in situ Li activation process, appearing two discharge plateaus at around 1.7 and 1.3 V on both Cu$_2$S@Cu and $Cu_2S_x$@Cu NRs, respectively. These plateaus correspond to the stepped conversion to form Li$_2$S and Cu (Supplementary Fig. 14). To gain a better understanding of the structural evolution during the process, in situ XRD characterization was performed. As shown in Fig. 2e (contour maps) and Supplementary Fig. 15 (one-dimensional maps, 1D maps), the diffraction peaks of $Cu_2S_x$ shifted to lower angles after reaching the two discharge plateaus on the GDC curve. A new phase of $Cu_{31}S_{16}$ appeared with enhanced peak intensities right after the first negative peak shift of $Cu_2S_x$. However, this phase subsequently disappeared after the second peak shift of $Cu_2S_x$, which has been reported as the electrochemical active intermediate[19]. Following these changes, the characteristic signals of cubic Li$_2$S rise up (JCPDS No.26-1188). Interestingly, the diffraction pattern was dominated by two prominent reflections at $2\theta$ of 26.9° and 55.7°, corresponding to the (111) and (222) facets of Li$_2$S. Simultaneously, Cu signals became more intensified and wider, indicating the formation of Cu clusters (Supplementary Fig. 16). TEM and HRTEM images along with the corresponding FFT diffractogram in Fig. 2f, g further confirm the microstructure of the Li$_2$S particles exposing (111) facet of cubic phase. Based on the FFT diffractogram, lattice fringes exposed to the surface were assigned to the (111) facet of Li$_2$S, as viewed along the <01$\bar{1}$> zone axis. To demonstrate the effect of introducing S defects, the electrochemical lithiation of Cu$_2$S@Cu was also monitored using in situ XRD technology with results shown in Supplementary Figs. 17–19 and Supplementary Note 4. In stark contrast to the evolution of $Cu_2S_x$@Cu, the Li$_2$S phase formed from Cu$_2$S@Cu did not show crystal surface selectivity, displaying significant reflections from (111), (222), (220) and (311) facets of Li$_2$S phase. This difference in Li$_2$S crystal facets strongly suggests that S defects play a crucial role in the formation of Li$_2$S(111)@Cu. To corroborate this hypothesis, we synthesized Cu$_2$S-200@Cu NRs with varying S defect content via tuning the annealing temperature of Cu$_2$S@Cu NRs to 200 °C. EPR was employed to demonstrate the reduction of S defect content in Cu$_2$S-200 compared to $Cu_2S_x$ (Supplementary Fig. 20). The specific S defect content of Cu$_2$S-200 was 8.6 atomic% from the XPS date and 7.1 atomic% from the EDS date, respectively (Supplementary Fig. 21). Following that, in situ XRD was conducted to track the Li activation process of Cu$_2$S-200@Cu, and the eventual material after Li activation was denoted as Li$_2$S-200@Cu. Unlike the lithiation of $Cu_2S_x$, the Li$_2$S phase formed from Cu$_2$S-200@Cu did not show crystal surface selectivity (Supplementary Fig. 22 and Supplementary Note 5). We calculated the intensity ratios of the reflections from (111), (220), (311), and (222) facets for Li$_2$S@Cu and Li$_2$S-200@Cu, yielding values of 1:1.31:0.70:1.74 and 1:0.28:0.23:0.25, respectively. As a side note, no obvious differences in facets were observed among the Cu clusters formed from Cu$_2$S and $Cu_2S_x$, or the Cu substrat (Supplementary Fig. 23 and Supplementary Note 6). In summary, Cu$_2$S-200@Cu with lower S defects could not

construct Li$_2$S(111)@Cu, however, relative to the Li$_2$S@Cu, the Li$_2$S-200@Cu exhibited an increased propensity to exhibit crystallographic facet selectivity. These results demonstrate the high impact of S defect on the Li$_2$S crystal facets.

To explore how S defects in $Cu_2S_x$ can induce the crystal selectivity in Li$_2$S, $Cu_2S_x$@Cu NRs were collected in the middle of lithiation process toward forming Li$_2$S(111)@Cu and analyzed by TEM. During the lithiation, defects including stacking faults (Supplementary Fig. 24a) and compositional variations were observed in the $Cu_2S_x$ region with S defects in $Cu_2S_x$@Cu NRs. We propose that the S defects facilitate the Li intercalation into the $Cu_2S_x$, forming Li$_y$CuS$_x$ and leading to a lattice expansion to generate stacking faults[20]. Compositional variations were verified by distinguished image contrast in HAADF-STEM/iDPC images and STEM-EELS, which shows Cu-deficiency in darker region and S-deficiency in brighter region (Supplementary Fig. 24b–e). In and above the Li$_y$CuS$_x$ region, crystalline Li$_x$Cu and amorphous phase, assumed to be Li$_x$S, were observed. The presence of Li$_x$Cu was confirmed by its diffusive reciprocal lattice points in FFT diffractogram obtained from HRTEM image, attributed to defective structures, contrasting with those from hcp LiCu nanoparticle. Further confirmation came from STEM-EELS spectra, which showed prominent signals for Li-K and Cu-M$_{2,3}$/L$_{2,3}$ edges, while S-L$_{2,3}$ edge signal was negligible (Supplementary 25). Although Li$_x$S could not be directly verified its amorphous nature is consistent with prior studies[21]. These observations along with the in situ XRD results suggest that S defects-driven Li intercalation might facilitate the diffusion of Cu and S from $Cu_2S_x$ region. This process could result in the replacement of Cu by S in the Cu-deficient region and S by Cu in the S-deficient region to form Li$_x$Cu, Cu, and potentially Li$_x$S species. As lithiation is further processed, we hypothesize that the amorphous Li$_x$S nanoparticles may transform into the cubic Li$_2$S nanoparticles predominantly exposing the thermodynamically most stable (111) facet[22]. The structural and compositional evolution occurred during the lithiation is summarized in Fig. 2h. In addition, the presence of defects could alter the arrangement of S sublattice[23], which may modify the reaction mechanism of Li$_2$S formation[24], thus potentially influencing the crystal orientation of Li$_2$S. Furthermore, online differential electrochemical mass spectrometry (OEMS) tests were carried out to gather more information on the electrolyte/electrode interface reactions. The results as shown in Supplementary Fig. 26 show the detection of CO$_2$ and C$_2$H$_4$ gases during the Li$_2$S@Cu activation, which were recognized as the products of electrolyte decomposition[25]. In contrast, no gas evolution occurred during the lithiation process in $Cu_2S_x$ NRs system, indicating the suppressed electron transfer from the electrode to the electrolyte[26]. Combinating with the results of XRD, EPR, XPS and XAS, the introduction of S defects tuned the electronic structure of electrode's surface, impacting the electron-transfer rate during the Li activation process.

## Effects of Li$_2$S(111)@Cu substrate on Li nucleation and growth behavior

To illustrate how the substrate affects Li deposition, varying amounts of Li were deposited on 3D Cu and Li$_2$S(111)@Cu substrates. As shown in Fig. 3a, b, the initial Li particles as highlighted in yellow with irregular-shapes show a random distribution (0.001 mAh cm$^{-2}$), with aggregative growth behavior becoming apparent at higher deposition levels (0.002 mAh cm$^{-2}$) on Cu substrate. As the Li deposition amount increases, SEM images as shown in Fig. 3c, d reveal a dendritic morphology and vertical orientation of Li (1 and 10 mAh cm$^{-2}$). This behavior is likely to be a consequence of the lithiophobicity nature of Cu substrate, characterized by a poor Li affinity and a large diffusion barrier (see Supplementary Fig. 27). In this case, the interaction between metal-to-substrate was lower than that of metal-to-metal, resulting in a "Volmer-Weber" or 3D island growth mode[15,27]. In contrast, on the Li$_2$S(111)@Cu NRs substrate, small and sphere-shaped Li

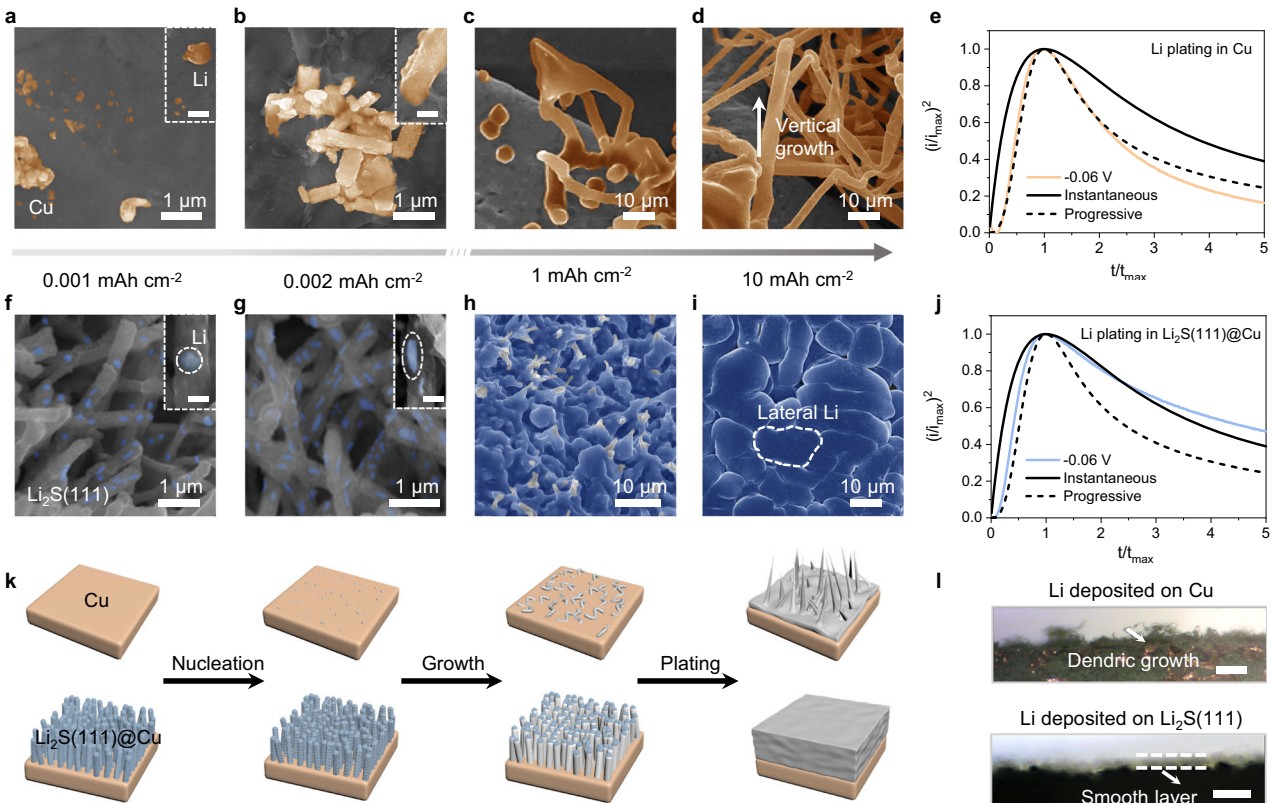

**Fig. 3 | Nucleation and growth of Li on various substrates.** SEM images of Li deposited on Cu (**a–d**) and Li$_2$S(111)@Cu (**f–i**) substrates with different capacities (the bar scale of the inset SEM images are 500 nm in (**a**, **b**) and 200 nm in (**f**, **g**), applied current density: 1 mA cm$^{-2}$). Dimensionless plots of current-time transient of Li deposition at −0.06 V on Cu (**e**) and Li$_2$S(111)@Cu (**j**) substrates (solid and dashed lines represent the theoretical graphs of 3D instantaneous and progressive nucleation-growth model, respectively). **k** Schematic illustration of Li deposited on Cu and Li$_2$S(111)@Cu substrates (the deposited Li are highlighted in gray). **l** In situ optical microscopy visualization of Li deposition (Applied current density: 2 mA cm$^{-2}$, time cutoff: 80 min, temperature: 25 ± 1 °C).

particles (highlighted by blue spots) were homogeneously distributed on the surface (0.001 mAh cm$^{-2}$), subsequently aggregating into droplet-shaped particles due to the rapid Li diffusivity (0.002 mAh cm$^{-2}$), as shown in Fig. 3f–i, Supplementary Figs. 28, 29 and Supplementary Note 7. Over time, the continued aggregation of deposited Li led to a liquid layer that completely enveloped the Li$_2$S(111)@Cu NRs (1 mAh cm$^{-2}$). Ultimately, the deposited Li filled up the spaces between Li$_2$S(111)@Cu NRs, resulting in a planar Li deposit (10 mAh cm$^{-2}$). Considering the high binding energy between Li$_2$S(111) and Li (Fig. 1c and Supplementary Fig. 3), the lateral Li morphology is depicted as the "Frank-van der Merwe" or layer-to-layer growth mode, where the metal-to-substrate attraction is stronger than that of metal-to-metal[26]. Noteworthy, the similar 3D vertical structure of Li$_2$S(111)@Cu, Li$_2$S-200@Cu and Li$_2$S@Cu NRs help to reduce local current density, and the high ion conductivity of Li$_2$S facilitates Li-transport, these are all advantageous for achieving uniform Li deposition. The process of Li deposition on Li$_2$S@Cu NRs substrate was characterized to explore the importance of exposed (111) plane of Li$_2$S for the lateral growth of Li. Unlike the dense Li deposition layer on Li$_2$S(111)@Cu NRs substrate, the lack of regularity in the crystal orientation on Li$_2$S@Cu NRs surface strongly affected the texture of Li deposits, resulting in a loose Li deposition layer on the outer surface of Li$_2$S@Cu NRs substrate (Supplementary Figs. 30, 31 and Supplementary Note 8). Subsequent investigations of Li deposition behaviors on these three substrates after first cycle revealed consistent phenomena (Supplementary Figs. 32–35 and Supplementary Note 9), demonstrating the impact of exposed (111) plane of Li$_2$S for the lateral growth of Li and the well-maintained effects of Li$_2$S(111)@Cu substrate. Further observations of the Li deposition morphology on the Li$_2$S-200@Cu NRs

exhibited a comparatively denser Li deposition than the Li$_2$S@Cu NRs substrate (Supplementary Figs. 36, 37 and Supplementary Note 10), which could be attributed to the increasing proportion of the (111) facet in Li$_2$S-200@Cu substrate. These experimental results align well with the previous DFT calculations, the latter have indicated that different crystal facets of Li$_2$S exhibit varying abilities to bind and transport Li, thus leading to different Li deposition behaviors on the various Li$_2$S-based substrates.

Electrochemical chronoamperometry (CA) measurements were conducted to elucidate the mechanism of Li nucleation and growth on various substrates. The current-time (*I-t*) transients of Li deposition on Cu and Li$_2$S(111)@Cu substrates at an overpotential of −0.06 V were recorded and are shown in Supplementary Fig. 38a, b, in which both current-time profiles exhibit a maximum current value ($I_m$) due to the two distinct effects: the enlargement of the surrounding growth surface and the overlap of the growth center[10], as explained by the Schariker and Hills (S-H) theory. The mathematical expressions of classical instantaneous and progressive nucleation are represented by the Scharifker equations (see Supplementary Note 11)[27]. By plotting $(I/I_m)^2$ *vs* t/t$_m$ and comparing it with Scharifker equations, the non-dimensional analysis of *I-t* transients of Li deposition on Cu and Li$_2$S(111)@Cu substrates with 3D instantaneous and progressive nucleation models are presented in Fig. 3e, j. For the Cu substrate, the behavior of Li deposition aligns well with 3D progressive nucleation model, which implies the presence of both new nuclei nucleation and pre-existing nuclei growth. Consequently, controlling the uniform deposition of Li becomes challenging, as illustrated in Fig. 3k. However, when considering the Li$_2$S(111)@Cu substrate, the initial Li nucleation process has two types of modes: progressive and

instantaneous nucleation. Notably, the portion after $t_{max}$ fits closely the instantaneous mode, which makes it easier to manipulate the final morphology of the deposition layer. The potential curves for Li plating in Supplementary Fig. 38c provide further evidence of the easier Li nucleation and growth on the Li$_2$S(111)@Cu substrate than those on the bare Cu substrate. Specifically, the nucleation ($R_n$) and growth resistances ($R_g$) of Li$_2$S(111)@Cu were 13 Ω and 7 Ω at 1 mA cm$^{-2}$, much lower than the 51 Ω and 25 Ω observed for the Cu substrate, respectively. As a result, the transition between nucleation and growth mode, combined with the reduced deposition resistance, all contributed to achieving smooth Li plating on Li$_2$S(111)@Cu substrate. Finally, these findings were visually confirmed using in situ optical microscopy technology, which recorded the evolution of Li morphology deposited on different substrates at a high current density of 3.5 mA cm$^{-2}$ as shown in Supplementary Figs. 39, 40 and Supplementary Movies 1, 2. In a specific case illustrated in Fig. 3l, a dense and flat layer of Li on Li$_2$S(111)@Cu surface was observed after 80 min of deposition. By contrast, the deposition of Li on bare Cu surface appeared uneven and fluffy, characterized by the presence of numerous Li whiskers. Additionally, the nucleation and growth of Li on Li$_2$S@Cu were also investigated to unveil dependency of Li deposition on different facets of Li$_2$S. As shown in Supplementary Fig. 41, Li deposition on Li$_2$S@Cu initially showed a similar behavior to that on Li$_2$S(111)@Cu substrates at initial nucleation ($t/t_{max} < 1$). However, the portion after $t_{max}$ deviated significantly from the instantaneous mode, which was considered as the impact of serious electrolyte decomposition[28]. Although the nucleation and growth resistances were slightly higher on the Li$_2$S@Cu substrate, as the deposition capacity increased, the Li deposition layer began to develop irregularities, accompanied by the formation of numerous Li dendrites on the sides (Supplementary Fig. 42 and Supplementary Movie 3). This means the 3D vertical array structure and lithiophilic characteristics of Li$_2$S alone may not be sufficient for Li to form a dense deposition. Instead, as the deposition layer thickened, Li tended to grow in the form of dendrites (Supplementary Fig. 43). Therefore, the behavior of substrate-dependent Li deposition appears to be influenced by the spatial structure, the nature of precursors, as well as the surface homogeneity, such as the crystal orientation of the substrates.

## Effects of Li$_2$S(111)@Cu substrate on Li plating/stripping behavior

So far, it has been demonstrated that the Li$_2$S(111)@Cu NRs substrate can influence the initial Li nucleation and growth behavior, and the regular crystal orientation of Li$_2$S ensures the dendrite-free morphology in the subsequent deposition stage. Accordingly, Li$_2$S(111)@Cu NRs could be considered as an ideal substrate for Li negative electrode materials. To assess the effect of the Li$_2$S(111)@Cu NRs substrate in battery systems, the Coulombic efficiency (CE) was first calculated using a partial-stripping mode, as proposed by Aurbach et al. [29]. As shown in Fig. 4a, the average CE based on Li$_2$S(111)@Cu substrate could reach 99.2% in an ether-based electrolyte at a current density of 1 mA cm$^{-2}$ with an areal capacity of 1 mAh cm$^{-2}$, while that of Cu substrate showed a lower average CE of 97.3% under the same measurement conditions. These CE values are repeatable (Supplementary Fig. 44). This implies that the Cu substrate experienced more active Li loss during cycling, possibly due to the Li dendritic growth, which led to a poor electrical connection[30]. To verify this hypothesis, the cell was disassembled after fully stripping the Li to examine the presence of inactive Li on substrates. As expected, SEM images as shown in the inset of Fig. 4a clearly reveal that almost no inactive Li could be found on the Li$_2$S(111)@Cu substrate, whereas dendrites and a substantial number of agglomerated particles were retained on the Cu substrate. In addition, the average CE of Li$_2$S@Cu substrate could also reach up to 99.0% under same conditions, where the SEM image showed limited inactive Li remained on the top of Li$_2$S@Cu NRs after stripping to the

cut-off potential (Supplementary Fig. 45). Therefore, to evaluate the Li plating/stripping reversibility on various substrates in a long-term cycling, the CEs tests were further measured under a plating-full-stripping mode. As anticipated, the cell based on Li$_2$S(111)@Cu substrate exhibited a good cyclability, consistently delivering stable CEs for over 500 cycles at 0.5 mA cm$^{-2}$ and 0.5 mAh cm$^{-2}$. However, the CE of the Cu substrate-based cell rapidly deteriorated after only 100 cycles under the same conditions (Fig. 4b). All these experiments can be repeatable and followed the same trend Supplementary Fig. 46. To explore the reason behind this marked difference in cyclability, SEM measurements were carried out to observe the structure of the cycled substrates. The morphology and structural integrity of Li$_2$S(111)@Cu substrate remained largely undisturbed after 50 cycles, whilst with the Cu substrate, its surface became rugged and appeared many large cracks (in the inset of Fig. 4b). Furthermore, notable disparities in lifespan and cycled morphology were also observed between the Li$_2$S(111)@Cu and Li$_2$S@Cu substrates. The cycle life of the Li$_2$S@Cu substrate-based cell is only half that of the Li$_2$S(111)@Cu substrate-based cell, and the vertical array structure of Li$_2$S@Cu NRs collapsed with some inactive Li accumulating on its surface after 50 cycles (Supplementary Fig. 47). These results align well with the morphological evolution of Li deposition in our previous discussion (Fig. 3f–i and Supplementary Figs. 30, 31), suggesting that the Li$_2$S@Cu substrate could not provide uniform Li deposition in the following cycles.

The CEs at a higher current density of 1 mA cm$^{-2}$ and a larger areal capacity of 1 mAh cm$^{-2}$ were also investigated to evaluate the Li diffusion kinetics. As expected, the Li$_2$S(111)@Cu substrate-based cell remained an enhanced stability, which could operate over 200 cycles, outperforming that of 60 cycles of Cu substrate (Supplementary Fig. 48). When increasing the current density and areal capacity to 3 mA cm$^{-2}$ and 3 mAh cm$^{-2}$, the Li$_2$S(111)@Cu substrate-based cell maintained stable cycling performance for 165 cycles. Whilst, the CE of the Cu substrate rapidly deteriorated only after 25 cycles (Supplementary Fig. 49). Upon further challenging test conditions, the Li$_2$S(111)@Cu substrate continued to demonstrate stability at 5 mAh cm$^{-2}$ and 5 mA cm$^{-2}$ (Supplementary Fig. 50).

To gain a deeper insight into the origin of the substrate-dependent Li deposition behavior, the interfacial charge transfer kinetics was examined. Firstly, Nyquist plots based on Li‖substrate half-cells were conducted both before and after CE tests (Fig. 4c, d and Supplementary Fig. 51). All impedance evaluations were summarized in Supplementary Table 5, where the Li$_2$S(111)@Cu substrate presented the smallest interfacial resistance compared with Li$_2$S@Cu and Cu substrates, indicating that more electrons could be supplied to the reduction of Li$^+$ at the nucleation stage. The corresponding linear sweep voltammetry (LSV) curves confirm the advantage of Li$_2$S(111)@Cu substrate in enhancing Li plating kinetics (Fig. 4e and Supplementary Fig. 52)[31]. After cycling, the electrolyte resistance ($R_e$) of the Li$_2$S(111)@Cu substrate cell remained almost unchanged, suggesting its stabilized interface. This observation aligns with the results from OEMS study as discussed above, which further supports the substrate's stability. The activation energy ($E_a$) for Li deposition on different substrates were calculated using temperature-dependent electrochemical impedance spectroscopy (EIS)[32]. Comparing the $E_a$ values, we found that the Li$_2$S(111)@Cu substrate had a lower Li deposition energy barrier at 53.46 kJ mol$^{-1}$ when compared to Cu (64.93 kJ mol$^{-1}$) and Li$_2$S@Cu (60.42 kJ mol$^{-1}$) (Fig. 4f–h and Supplementary Fig. 53), suggesting that the Li$_2$S(111)@Cu substrate exhibited a good lithiophilic character. It should be noted that the effects of Li$_2$S-200@Cu substrate on Li plating/stripping behavior were also evaluated, as the results shown in Supplementary Figs. 54, 56, the Li$_2$S-200@Cu demonstrated faster Li plating kinetics, higher Li utilization and improved Li plating/stripping reversibility compared to the Li$_2$S@Cu substrate, while it remained inferior to the Li$_2$S(111)@Cu substrate. Taking all these results into consideration, the Li$_2$S(111)@Cu substrate has the ability to

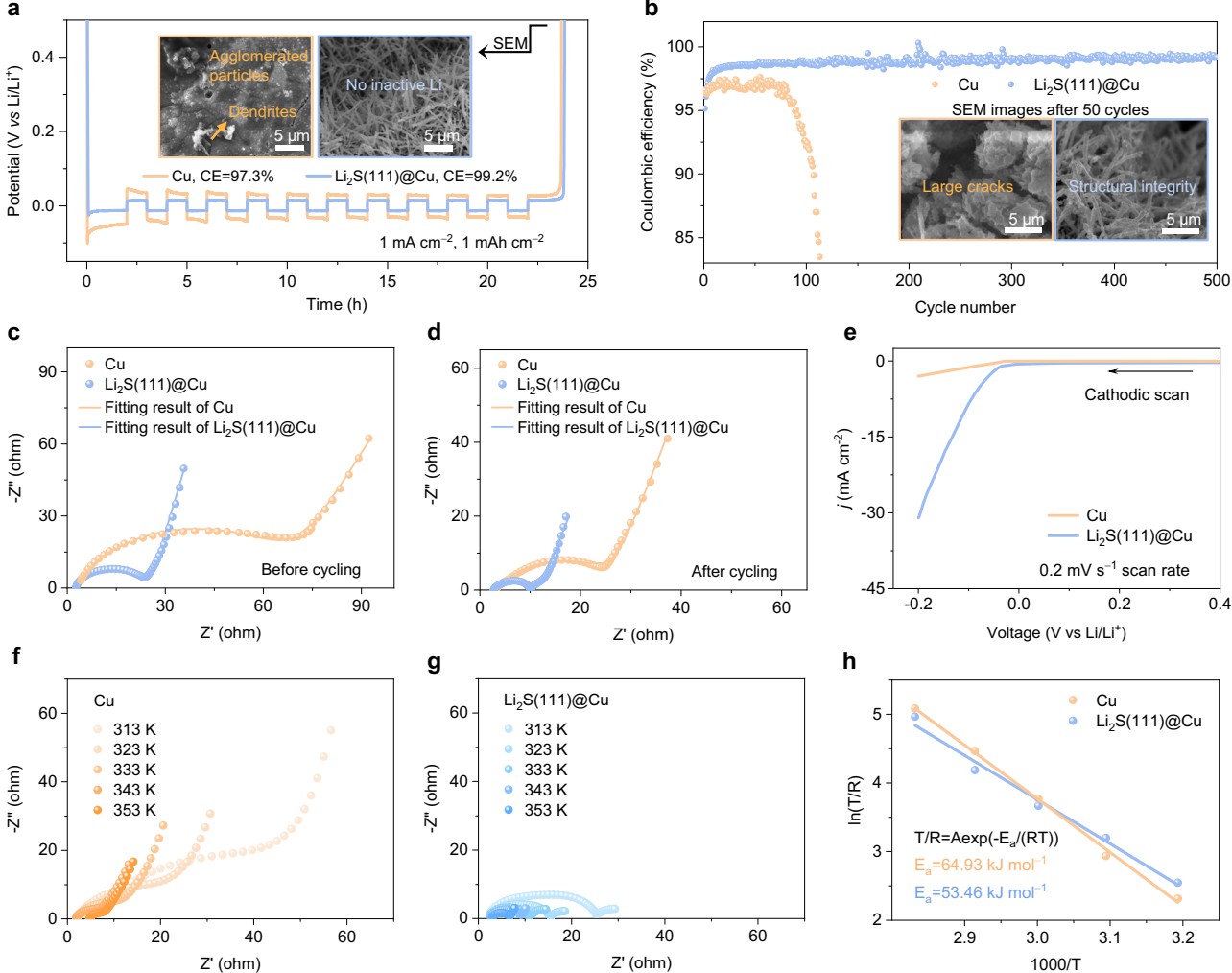

**Fig. 4 | Electrochemical performance of Li‖substrate half-cells with Cu and Li₂S(111)@Cu substrates. a** Potential profiles of CE test under a partial-stripping mode, the insets are SEM images of the substrates to detect inactive Li residue. **b** CE versus cycle number under a plating-full-stripping mode at 0.5 mA cm⁻² with 0.5 mAh cm⁻², the insets are SEM images of the substrates after 50 cycles. **c**, **d** EIS spectra before and after CE test in (**a**). **e** LSV curves at a scan rate of 0.2 mV s⁻¹. **f**, **g** EIS spectra obtained at different temperatures. **h** Arrhenius plots with activation energy obtained on each substrate.

transform the Li deposition behavior due to the lithiophilic nature of Li₂S and its homogeneous (111) facet orientation. This transformation enables a uniform Li nuclei distribution, facilitates interfacial charge transfer, and enhances Li diffusion in the electrolyte.

## Electrochemical performance of the symmetric and full cells

To further extend the effect of the Li₂S(111)@Cu substrate on inhibiting Li dendrite growth, the Li‖Li symmetric cells were assembled and tested by employing various substrates with less Li pre-deposit. As shown in Fig. 5a, the cell with Cu substrate exhibited a noticeable potential drop after depositing Li for 10 h at 0.5 mA cm⁻². Subsequently, the unrestrained Li dendrite growth led to a further drop in deposition potential, ultimately causing the cell to collapse after ~130 h. In contrast, the cell with Li₂S@Cu substrate maintained a steady deposition potential for ~700 h until a short circuit behavior occurred (Supplementary Fig. 57) thanks to the 3D vertically arrayed structure, which provided more space for Li deposition. Most notably, the Li₂S(111)@Cu substrate allowed for an extended duration of Li deposition, lasting ~800 h (Fig. 5a). This underscores the positive effect of the homogenously lithiophilic Li₂S(111) surface in regulating smooth Li deposition. Subsequently, the rate performance and Li plating/stripping stability of symmetric cells with 5 mAh cm⁻² of Li pre-deposited on the three substrates were investigated at a range of

current densities, 0.5, 1, 2, 3, and 4 mA cm⁻² with a fixed capacity of 1 mAh cm⁻². The cell based on Li₂S(111)@Cu substrate expressed the lowest overpotential of 12, 18, 31, 44, and 56 mV for the respective current densities compared to that of cells using Li₂S@Cu and Cu substrates (Fig. 5b and Supplementary Fig. 58). Conversely, the overpotentials of the Cu substrate sharply increased, and the potential curves even became asymmetric at high current densities, indicating poor interface stability and sluggish mass-transfer kinetics[33]. The exchange current densities ($j_0$) were obtained from the Tafel plots to quantify the Li transfer process. As shown in Supplementary Fig. 59, the $j_0$ of the cell based on Cu substrate (1.04 mA cm⁻²) is the lowest, indicating the sluggish mass-transfer kinetics and the highest overpotentials. In contrast, Li₂S@Cu and Li₂S(111)@Cu substrates exhibited higher $j_0$ values (1.75 mA cm⁻² and 2.25 mA cm⁻², respectively), demonstrating faster and more reversible Li plating/stripping. In addition, the enhanced Li transport kinetics was also substantiated by the improved Li transference number and ionic conductivity when using the Li₂S(111)@Cu substrate (Supplementary Figs. 60, 61 and Supplementary Table 6). Furthermore, the Li₂S(111)@Cu substrate can operate stably for more than 1500 h when the current density returned to 0.5 mA cm⁻² from the high current density of 4 mA cm⁻², where the cells based on Li₂S@Cu and Cu substrates exhibited 800 and 460 h, respectively.

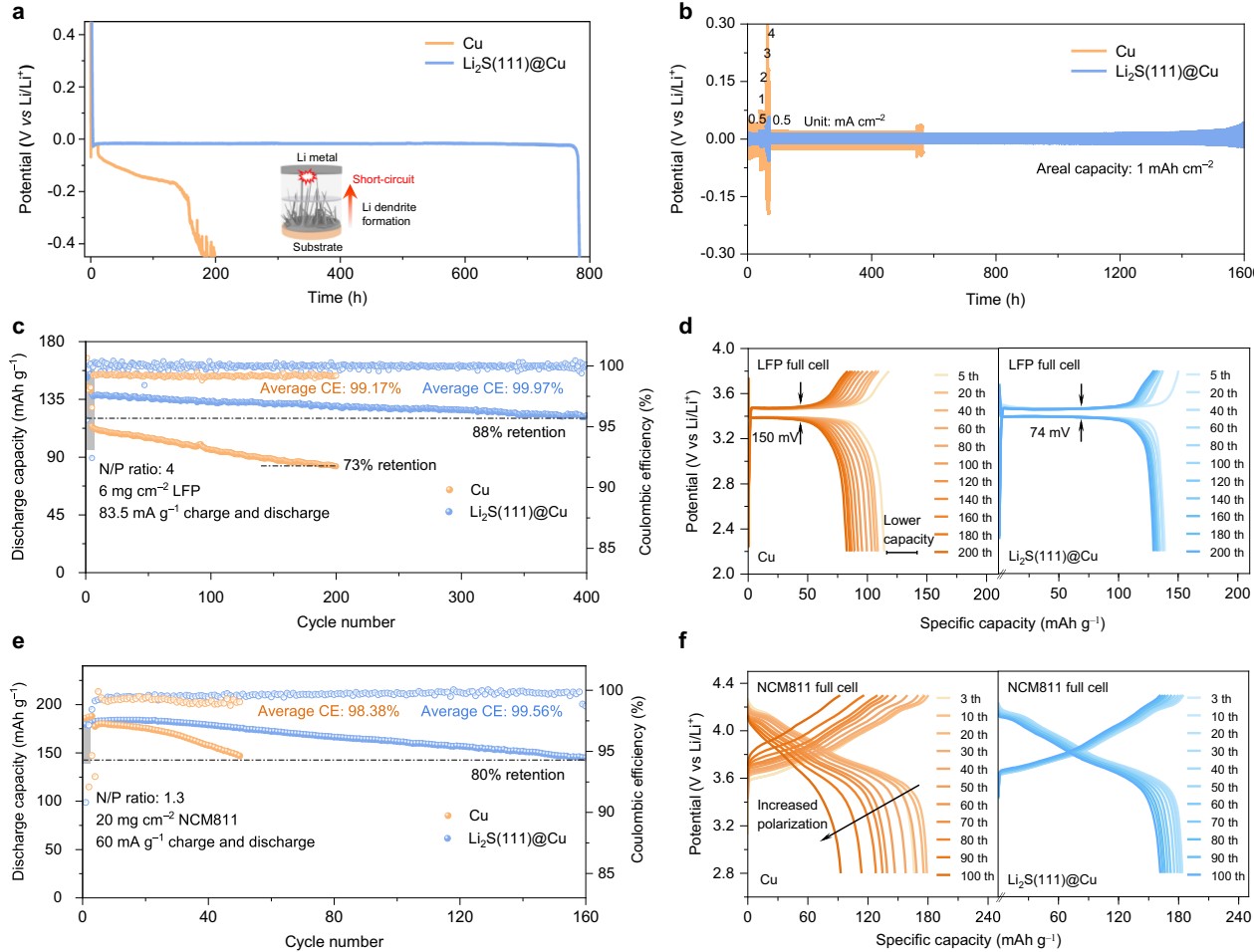

**Fig. 5 | Electrochemical performance of symmetric and full cells based on Cu and $Li_2S(111)@Cu$ substrates. a** Potential-time curves of short-circuit test for Li‖substrate cells at 0.5 mA cm⁻². **b** Rate performance of the symmetric cells at different current densities from 0.5 to 4 mA cm⁻² with an areal capacity of 1 mAh cm⁻². **c** Discharge capacity retention of Li-substrate ‖LFP full cells at 83.5 mA g⁻¹. **d** Corresponding GDC profiles of LFP full cells at different cycles. **e** Discharge capacity retention of Li-substrate ‖NCM811 full cells at 60 mA g⁻¹. **f** Corresponding GDC profiles of NCM811 full cells at different cycles.

To demonstrate the practical application of our research, we assembled full cells using negative electrodes composed of $Li_2S(111)$@Cu nanorod substrates with pre-deposited Li, and positive electrodes made of a lithium iron phosphate material ($LiFePO_4$, named as LFP) or a Ni-rich layered material ($LiNi_{0.8}Co_{0.1}Mn_{0.1}O_2$, named as NMC811) with a high areal loading of 6 or 20 mg cm⁻². As shown in Fig. 5c, the full cell based on $Li_2S(111)$@Cu negative electrode substrate and LFP positive electrode displayed stable cyclability with a high capacity retention of 88% after 400 cycles at 83.5 mA g⁻¹. In contrast, when using the Cu or $Li_2S$@Cu as negative electrode substrate (Supplementary Fig. 62a), it exhibited a lower discharge capacity and average CE under the same conditions. In addition, the polarization potential value of $Li_2S(111)$@Cu based full cell (74 mV) is lower than the Cu (150 mV) and $Li_2S$@Cu (83 mV) based full cells, as observed in Fig. 5d and Supplementary Fig. 62b, indicating a faster Li transport dynamics. Therefore, we further conducted the full cell testing at the higher current densities of 500 mA g⁻¹ and 835 mA g⁻¹, the result as shown in Supplementary Fig. 63 and Supplementary Note 12. The LFP full cells based on the $Li_2S(111)$@Cu substrate maintained steady capacities, with 80% capacity retentions over 1000 cycles (500 mA g⁻¹) and 2000 cycles (835 mA g⁻¹). However, the full cells based on Cu substrate exhibited pronounced deteriorations, with 80% capacity retentions only after 54 cycles (500 mA g⁻¹) and 62 cycles (835 mA g⁻¹). Further evaluation of $Li_2S(111)$@Cu as negative electrode substrate was conducted by constructing full cells using commercial NMC811 positive

electrodes. As shown in Fig. 5e, the full cell based on Cu as negative electrode substrate and NCM811 as positive electrode displayed a rapid degradation with only 80% capacity retention over 49 cycles at 60 mA g⁻¹. By contrary, with the same NCM811 positive electrode but $Li_2S(111)$@Cu based full cell exhibited better performance, delivered higher capacities and extended longevity by over threefold, it maintained stable cycling for 160 cycles. In addition, the initial CE of Cu based full cell was only 80.3%, which is much lower than that of 90.8% observed for the $Li_2S(111)$@Cu based full cell. Furthermore, the average CEs of Cu based full cell was around 98.4% within 80% capacity retention, while that of $Li_2S(111)$@Cu based full cell reached up to 99.6%. To elucidate the reason behind the notably different performance, the charge/discharge profiles of various cycles are presented in Fig. 5f, where a substantial increase of potential polarization was observed in the Cu based full cell over cycling, indicating the massive loss of active Li, primarily due to the growth of Li dendrites. Conversely, no significant potential drop was observed in the $Li_2S(111)$@Cu based full cell, attributed to its improved interface stability resulted from even Li plating/stripping. All these experiments with commercial NMC811 positive electrodes are repeatable when assembled with $Li_2S(111)$@Cu and Cu substrates (Supplementary Fig. 64). The performance of $Li_2S$@Cu and $Li_2S$-200@Cu based NCM811 full cells were also examined; which exhibited moderate lifespans of 86 and 107 cycles with 80% capacity retention, respectively (Supplementary Figs. 65, 66). Finally, the high-rate tests of the $Li_2S(111)$@Cu substrate in the NCM811

full cell were tested, the results as shown in Supplementary Fig. 67. The full cell based on the Cu and Li$_2$S(111)@Cu substrate delivered an initial capacities of 107.2 and 159.5 mAh g$^{-1}$ at 600 mA g$^{-1}$, and presented capacity retentions of 80% after 17 and 163 cycles, respectively (Supplementary Fig. 67a). Even at 1000 mA g$^{-1}$, the Li$_2$S(111)@Cu substrate continuedly presented considerable stability, with 80% capacity retention after 177 cycles and the high average CE of 99.1% (Supplementary Fig. 67c). However, the Cu based NCM811 full cell deteriorated quickly, with capacity dropping from 76.1 to 61 mAh g$^{-1}$ (80% retention) within only 7 cycles, and the average CE as low as 69.0%. The corresponding charge/discharge profiles in Supplementary Fig. 67b, d further illustrated the rapid capacity decay and large potential polarization for the Cu substrate, indicating the uneven Li plating/stripping and sluggish Li transfer kinetics. The performance of the Li$_2$S(111)@Cu substrate, as observed at high current densities within the LFP and NCM811 full cells, suggesting its potential for high-rate applications. The performance of full cells based on Li$_2$S(111)@Cu substrate surpassed that of most state-of-the-art optimized substrates in the literature (refer to Supplementary Table 7). Lastly, pouch cell were employed to demonstrate the application potential of the Li$_2$S(111)@Cu substrate. As shown in Supplementary Fig. 68, the pouch cell fabricated by high-loading LFP positive electrode also exhibited a maximum discharge capacity of 99.96 mAh at 55 mA g$^{-1}$, and maintained a capacity retention of 83% after 200 cycles with the average CE of 99.9%. The morphology of the cycled Li$_2$S(111)@Cu substrate was displayed in Supplementary Fig. 69, demonstrating good morphology integrity.

## Discussion

The crystal orientation of Li$_2$S substrate is discovered to be important for Li deposition behavior on developing high-performance lithium metal batteries (LMBs). The close-packed Li$_2$S(111) facet is favorable for achieving dense Li deposition due to its high Li affinity and low diffusion barrier, in comparing to open and stepped facets of Li$_2$S(110) and Li$_2$S(311). Thanks to the uniform distribution of Li nuclei and fast electron/ion transport at Li/substrate interface, the Li$_2$S(111)@Cu substrate displayed a mixed Li nucleation mode during initial deposition followed by a mono-instantaneous growth mode, resulting in a smooth Li layer deposited on Li$_2$S(111)@Cu substrate. Finally, both Li||Cu half cells and Li||Li symmetric cells constructed using the Li$_2$S(111)@Cu substrate show remarkable improvements in terms of Coulombic efficiency (CE) and cyclability. This work reveals the impact of electrode substrate design and engineering on Li nucleation and growth behavior, for achieving a dendrite-free Li deposition that is essential for safe and durable operation of LMBs and beyond.

## Methods

### Materials

Cu foam (purity 99.9%) was purchased from Guangzhou Bojing Technology Co., Ltd. Hydrochloric acid (HCl), ammonium persulfate ((NH$_4$)$_2$S$_2$O$_8$, ≥98%), sodium hydroxide (NaOH, ≥96%) and sodium sulfide nonahydrate (Na$_2$S·9H$_2$O, ≥98%) were purchased from Sinopharm Co., Ltd and used as received. The deionized water (DI, 18.2 MΩ·cm) used throughout synthetic process was generated from a Milli-Q system (Nihon Millipore Ltd). The LFP and Super P materials were purchased from Guangdong Canrd New Energy Technology Co., Limited. The polyvinylidene difluoride (PVDF, molecular weight:1000,000, purity 99.5%) and N-methyl-2-pyrrolidone (NMP, purity 99.9%) were purchased from Canrud. The carbon@Al foil (15 μm, purity 99.9%) was purchased from Shenzhen Kejing, used as received. The high areal loading NCM811 positive electrode electrodes (4.0 mAh cm$^{-2}$, thickness of 70 μm) were provided by CATL (Contemporary Amperex Technology Co., Limited). The NCM811 active material in the composite positive electrodes is 95 wt%. The lithium (Li) foils (16 mm in diameter, purity 99.9%) were purchased from CEL

(China energy lithium Co., Limited). The electrolyte consisted of 1 M lithium bis(trifluoromethanesulfonyl)imide (LiTFSI) and 2 wt% LiNO$_3$ in dimethoxyethane (DME) and 1,3-dioxolane (DOL) with a volume ratio of 1:1 was purchased from Dodochem (Suzhou, China), denoted as LS-009 electrolyte. The electrolytes were prepared in an Ar-filled glove box (O$_2$ < 0.1 ppm, H$_2$O < 0.1 ppm). Ethylene carbonate (EC, purity 99.95%), diethyl carbonate (DEC, purity 99.99%), fluoroethylene carbonate (FEC, purity 99.9%), and LiPF$_6$ (purity 99.9%) were purchased from DoDoChem and used as received. The CR2032 case and spring (material: 304 stainless steel) and the Celgard 2325 separator (thickness: $25 \pm 2$ μm, lateral dimension: $100 \pm 0.2$ mm, porosity: $39 \pm 5$%, average pore diameter: 0.06 μm) were purchased from Canrud, and used as received.

### Synthesis of Li$_2$S@Cu substrate

Firstly, a piece of Cu foam was washed by using 1 M HCl to remove surface impurities, and then rinsed with DI water. Subsequently, the Cu foam was immersed into 20 mL DI water containing 2.0 g NaOH and 0.43 g (NH$_4$)$_2$S$_2$O$_8$ to generate Cu(OH)$_2$@Cu foam. After rinsing with DI water, the sample was sulfurized through hydrothermal reaction at 100 °C for 6 h in a 40 ml solution containing 0.4 g Na$_2$S·9 H$_2$O to obtain Cu$_2$S@Cu foam. The latter was rinsed with DI water several times and vacuum dried at 60 °C for 12 h. The prepared Cu$_2$S@Cu foam was cut into a disk of 16 mm in diameter and assembled into the Li||Cu$_2$S@Cu half cells, after a lithium activation process (the cells were discharge at 0.5 mA cm$^{-2}$ to 0 V, the electrolyte used was LS-009 electrolyte, with a volume of 60 μL), the Li$_2$S@Cu substrate was formed.

### Synthesis of Li$_2$S(111)@Cu substrate

The Cu$_2$S@Cu foam was placed in the middle of a quartz tube in the CVD furnace. The tube was first ventilated with Ar (400 sccm) for 15 min to remove impurities and create an inert atmosphere, and then kept under Ar (40 sccm) flow in the tube during the whole fabrication process. After that, the furnace was heated to 280 °C within 25 min and kept at the specific temperature for 60 min, Cu$_2$S$_x$@Cu foam was prepared. During the annealing process, the migration of sulfur atoms occurs due to the thermodynamic drive, resulting in distortions of the periodic structure in crystal, thereby generating sulfur defects[34]. Finally, the Li$_2$S(111)@Cu substrate was in situ constructed by the electrochemical lithiation process.

### Characterizations

The morphologies of the samples were characterized by SEM (Hitachi S-4800, Hitachi, Japan), TEM and EDS (FEI Tecnai 30, USA). Double Cs-corrected (S)TEM systems (Themis Z, Thermo-Fisher Scientific) equipped with EELS (Quantum ER965, Gatan) and EDS (Super-X EDS system) were used for atomic-scale structure imaging and chemical analysis of the samples at an accelerating voltage of 300 kV. Due to the electron beam damage of Li$_2$S sample by high-energy electron illumination in TEM, we acquired high-resolution HAADF and iDPC-STEM images, and EELS and EDS data at the low dose rates. The chemical environment and bonding information were detected by XPS (PHOI-BOS150, Germany). When the samples were collected from cells, the cycled cells were disassembled in the Ar-filled glove box. The electrodes were rinsed with DOL (for half cells) or DEC (for full cells) three times, and then dried at transition cabinet under vacuum for overnight. When the samples to be characterized contains Li, a vacuum transfer chamber was implemented for transferring the samples from glovebox into SEM or XPS chambers to avoid degradation toward air. The crystalline structures were investigated by XRD with Cu-Kα radiation (Rigaku Ultima IV). The EPR measurements were carried out with Bruker ESR300E at low temperature.

## XAS measurements and analysis

X-ray absorption fine structure (XAFS) spectra at Cu K-edge were obtained on the 1W1B beamline of Beijing Synchrotron Radiation Facility (BSRF) operated at 2.5 GeV and 250 mA. To avoid the signal interference of Cu foam, $Cu_2S$, and $Cu_2S_x$ samples were prepared by collecting the powder obtained from ultrasonically treated $Cu_2S@Cu$ and $Cu_2S_x@Cu$ substrates. The Athena software (version 0.9.26) was employed to calibrate the background, pre-edge, and post-edge lines. The coordination number (CN), bond length (R), Debye-Waller factor ($\sigma^2$), and $E_0$ shift ($\Delta E_0$) were obtained after EXAFS fitting. The Cu foil fitting parameters: $k^3$ weighting, k-range (3–12 $\text{Å}^{-1}$) and R range (1–3 Å); the $Cu_2S$ and $Cu_2S_x$ fitting parameters: k-range (3–9 $\text{Å}^{-1}$) and R range (1–3 Å). WT analysis was employed to visualize the fitting results, the mother wavelet was Morlet, the parameters were: $k^3$ weighting, k-range (0–15 $\text{Å}^{-1}$) and R range (1–4 Å).

## In situ XRD measurements

In situ XRD patterns were collected using a Bruker D8 A25 diffractometer (Germany, λ¼1.5418 Å) with Cu Kα radiation. The in situ cells were made from CR2025-type coin cell casings, a 6 mm diameter hole was drilled in the center and sealed with Kapton polyimide film (9 mm in diameter) using Torr Seal epoxy.

## OEMS measurements

The OEMS experiments were carried out using an Agilent mass spectrometer and a custom-designed electrochemical flow cell. The mass-selective detector was a modified 5975 C, the carrier gas was Helium and the flow rate was maintained at 5 mL $\text{min}^{-1}$ during testing.

## In situ optical microscopy measurements

Real-time optical microscopy was conducted using a three-eye stereomicroscope (SG900, Suzhou Shenying Optical Instrument Co., Ltd) and home-made electrochemical optical cell.

## Electrochemical tests

CR2032-type coin cells were prepared in an Ar-filled glove box. One spring (15.5 × 1.1 mm) and One gasket (15.8 × 1.0 mm) were used in each cell. The Li‖Cu half cells based on different substrates were assembled with Li foil (500 μm in thickness, 16 mm in diameter) as the negative electrode and the Celgard 2325 (19 mm in diameter, one piece for each cell) as the separator. The applied electrolytes were 1 M LiTFSI dissolved in a mixture of DOL and DME in a volume of 1:1 with 2 wt% LiNO₃ or 1 M LiPF₆ dissolved in a mixture of EC and DEC in a volume of 1:1 with 5 wt% FEC. The volume of electrolyte used in Li‖Cu half cells was 60 μL. The CE data of Li‖Cu half cells based on a partial-stripping mode which proposed by Aurbach et al. can be calculated by the following equation[29]:

$$CE = \frac{nQ_C + Q_r}{nQ_C + Q_l} \qquad (1)$$

where $Q_c$, $Q_r$ and $Q_l$ correspond to the charges involved in a single plating/stripping process, final charging (the residual Li) and initial loading of Li, respectively. Herein, the $Q_c = 1$ mAh $\text{cm}^{-2}$, $Q_l = 2$ mAh $\text{cm}^{-2}$, $n = 10$ (cycles). The Li‖Li symmetric cells were based on different substrates with 5 mAh $\text{cm}^{-2}$ Li pre-deposit and 60 μL electrolyte. During the Li pre-deposition, the Li‖substrate half cells were assembled, and then a fixed amount of Li was deposited under 0.5 mA $\text{cm}^{-2}$ with 60 μL LS-009 electrolyte. The LFP positive electrodes were prepared by mixing the LFP, Super P and PVDF in a weight ratio of 9:1:1 in NMP, then stirring for 6 h to form a homogeneous electrode slurry. The slurry was pasted on one side of the C@Al foil (15 μm, Shenzhen Kejing, used as received) through a doctor blade (250 μm), and dried at 110 °C for 12 h. The LFP positive electrodes were obtained by punching the dried foil into circular discs with a diameter of 12 mm

using the MAK-T10 slicer (Shenzhen Kejing). The LFP full cells were assembled with LFP positive electrodes (-6 mg $\text{cm}^{-2}$, 12 mm in diameter), 1 M LiTFSI dissolved in a mixture of DOL and DME in a volume of 1:1 with 2 wt% LiNO₃ as the electrolyte, with a volume of 30 μL, and the potential range was 2.2–3.8 V. The LFP pouch cell (100 mAh, 7.5 × 7.5 $\text{cm}^2$) was assembled with high-loading LFP (20 mg $\text{cm}^{-2}$), the electrolyte was injected through a syringe (3.3 g $\text{Ah}^{-1}$), and the potential range was 2.5–3.85 V. After formation process (charge to 4 V at 8 mA $\text{g}^{-1}$ and discharge to 2.5 V at 55 mA $\text{g}^{-1}$), the gas bag was cut open to release gases, followed by vacuum sealing. The external pressure applied during the cycling of LFP pouch cell was 37.5 kPa. The mass proportion of each component in the pouch cell was listed in Supplementary Table 8. The NCM811 full cells were assembled with NCM811 positive electrodes (20 mg $\text{cm}^{-2}$, 12 mm in diameter), different substrates with pre-deposited Li as negative electrodes, 1 M LiPF₆ dissolved in a mixture of EC and DEC in a volume of 1:1 with 5 wt% FEC as the electrolyte, and the potential range was 2.8–4.3 V. The electrolyte amount was 7.5 μL $\text{mAh}^{-1}$. All cells were tested using the LANDHE CT2001A system or Neware battery CT-4008 testing system under galvanostatic charge/discharge mode in the thermostatic chamber at 25 ± 1 °C.

LSV was performed in a three-electrode cell with a sweep speed of 0.2 mV $\text{s}^{-1}$ on an electrochemical workstation (CHI760E, CH Instruments, USA). Li metal electrodes with an area of 2 $\text{cm}^2$ were used as the reference and counter electrodes. The Li nucleation and growth mode was investigated using chronoamperometry analysis for the Li‖substrate cells, the step potential was fixed at -0.06 V. EIS tests were carried out with a frequency range of $10^5$ Hz–0.1 Hz and an amplitude of 5 mV. The EIS tests were conducted under open-circuit potential, which was obtained by appling the open-circuit voltage time for 10 min. During the EIS test, 12 points were collected per decade of frequency. The temperature- dependent EIS spectra for Li‖substrate cells were collected to compare the deposition energy barrier, which can be calculated by the following equation[32]:

$$\frac{T}{R_{ct}} = A exp\left(-\frac{E_a}{RT}\right) \qquad (2)$$

where $E_a$ is the activation energy, T is the absolute temperature, $R_{ct}$ is the interfacial resistance, A is the pre-exponential factor, and R is the gas constant.

The substrates with 10 mAh $\text{cm}^{-2}$ Li pre-deposit were employed to prepare the cells to evaluate the mass transfer kinetic, the exchange current density was obtained from the Tafel plots (at 2 mV $\text{s}^{-1}$ scan rate in the potential range from −200 mV to +200 mV) fitted with the Bulter–Volmer equation[35]:

$$\eta = a + b log j \qquad (3)$$

where η represents the overpotential, the j₀ can be calculated from the intersection of the extrapolated linear part of the log j versus η plot with the η = 0.

The transference number of Li⁺ was determined in the Li‖Li symmetric cells with various substrates. The chronoamperometry curves were recorded under a polarization potential of 10 mV for 5000 s. Meanwhile, the EIS tests before and after polarization were carried out with a frequency range of 105 Hz−0.1 Hz and an amplitude of 5 mV. t+ was calculated using the following equation[36]:

$$t^+ = \frac{I_{ss}}{I_0}\left(\frac{\Delta V - I_0 R_0}{\Delta V - I_{ss} R_{ss}}\right) \qquad (4)$$

where ΔV was the applied polarization potential; $I_0$ and $R_0$ were the initial current and interfacial resistance, respectively; $I_{ss}$ and $R_{ss}$ were the steady-state current and interfacial resistance, respectively.

## Theoretical calculations

The DFT calculations were performed using the Vienna Ab-initio Simulation Package (VASP) code for investigating the adsorption properties and migration behaviors. The exchange-correlation was described with the generalized gradient approximation (GGA) and the Perdew-Burke-Ernzerhof (PBE) function. The electronic analysis was based on the DDEC6 approach[11]. The CI-NEB method was carried out to calculate the energy barrier[10]. The convergence tolerance was reached when the energy change was smaller than $10^{-6}$ eV, and 0.01 eV Å$^{-1}$ for maximum residual force. The spin polarization was considered in all calculations. During the process of geometry optimization, a kinetic energy cutoff of 500 eV was used. To avoid the interaction caused by the periodicity, a vacuum layer of 14 Å along the z-direction was used for calculations. The atomic layers were 7, and the supercells of $Li_2S$ (311), (110) and (111) were ($1 \times 1 \times 2$), ($2 \times 2 \times 1$) and ($2 \times 2 \times 1$), respectively. The optimized lattice constants of $Li_2S$(311) were a = 13.49 Å, b = 26.46 Å, c = 4.14 Å, α = β = γ = 90°; (110): a = 11.39 Å, b = 8.06 Å, c = 26.09 Å, α = β = γ = 90°; and (111): a = b = 8.03 Å, α = β = 90°, γ = 120°. In the calculated configurations presented in figures, Li, S, and adsorbed Li atoms were represented by gray, blue, and pink, respectively. The $E_{binding}$ was defined as follows:

$$E_{binding} = E_{slab+Li} - E_{slab} - E_{Li} \qquad (5)$$

where the $E_{slab+Li}$ represented the total energy after adsorption, $E_{slab}$ and $E_{Li}$ represented the energy of isolated slab and Li, respectively.

## Data availability

The data supporting the findings in this study are present in the paper and the Supplementary Information files. Source data are provided with this paper. The raw data used in this study are available in the figshare database under accession code: https://doi.org/10.6084/m9.figshare.28327538. Source data are provided with this paper.

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

## Acknowledgements

This work was supported by the National Key Research and Development Program (No. 2021YFB2400300, No. 2022YFB2502103) and Natural Science Foundation of China (grant numbers 22172133, 22288102). G.-S.P. was supported by the National Research Foundation of Korea (NRF) grant by the Ministry of Science, ICT & Future Planning (MSIP) (No. RS-2023-00258732). S.Z. thanks the support from Natural Science Foundation of China (grant numbers 223B2905). W.-F.L. thanks the support from the EPSRC (EP/W03784X/1). We are thankful to the Beijing Synchrotron Radiation Facility (1W1B, BSRF) for their help with the characterizations.

## Author contributions

S.-G.S. and L.H. contributed to the conception of the study. J.-X.L. designed the experiments, characterizations, electrochemistry and wrote the paper. R.O., G.-S.P., X.-Y.H., W.-F.L., L.H., and S.-G.S. contributed significantly to the data analysis and manuscript revision. R.O. and G.-S.P. performed TEM characterizations. P.D. conducted the in situ XRD, OEMS, and in situ optical microscopy experiments. S.-N. H. conducted the materials synthesis, TEM, XAS and contacted DFT calculations. P.D., S.-N.H., S.Z., C.-G.S., and J.-F.S. helped perform the analysis with discussions. Y.-X.X., W.-C.Z., H.C., S.-S. L., H.-Y.H, Y.Z., and J.-T.L. participated in data analysis. The project was supervised by L.H. and S.-G.S. All authors contributed to the discussion of the results.

## Competing interests

The authors declare no competing interests.
