## [Peer Review File · Nature Communications]

Sulfur defect engineering controls Li₂S crystal orientation towards dendrite-free lithium metal batteries

Corresponding Author: Professor Shi-Gang Sun

Version 0:

Reviewer comments:

Reviewer #1

(Remarks to the Author)

The manuscript investigates the important of crystal orientation of Li₂S substrate for Li deposition behavior on developing high-performance lithium metal batteries. The author used DFT calculations to predict the close packed Li₂S(111) facet is favorable for achieving dense Li deposition. Two assemblies of the Li₂S(111)@Cu anode substrate with LiFePO₄ cathode that retained 88% capacity after 400 cycles, and with a commercial LiNi_{0.8}Co_{0.1}Mn_{0.1}O₂ cathode (4 mAh cm⁻²) that showcased 160 cycles. However, the manuscript is lack of innovations. Therefore, this manuscript is not recommended to be published in Nature Communication.

1. What is the amount of sulfur defects in Cu₂Sx? How to regulate the content of sulfur defects. How does the sulfur defect content regulate the Li₂S crystal plane and how does it affect the Li deposition process?
2. How to ensure the morphology and uniformity of Li₂S deposition during the conversion process between Cu₂Sx and Li₂S?
3. Generally, Li₂S has poor electronic conductivity. After depositing a layer of Li₂S on the Cu current collector, how can it promote the occurrence of Li deposition electrochemical reactions? How does Li₂S regulate the deposition behavior of lithium when lithium deposition covers the Li₂S layer?
4. Is there a basis for the reaction mechanism of sulfur vacancy regulation of electronic structure changes in Page 10? The author should elaborate the changes in the mechanism in detail.
5. Is a sudden drop in voltage in Fig.5b a short circuit? The author needs to provide relevant data.

(Remarks on code availability)

Reviewer #2

(Remarks to the Author)

The submission by Sun et al. reported a three-dimensional Li₂S(111)@Cu nanorod array as the Li deposition substrate through sulfur defect engineering for Li₂S crystal orientation control, which exhibited efficient regulation of Li-ion adsorption, nucleation, and deposition. The authors investigated the interactions between Li and various surface structures of Li₂S, and revealed that Li₂S(111) plane exhibits the highest Li affinity and lowest diffusion barrier, resulting in dense Li deposition. The Li₂S(111)@Cu substrate showed a mixed Li nucleation mode during initial deposition followed by a mono-instantaneous growth mode, which facilitated smooth Li deposition. The Li₂S(111)@Cu anode substrates can pair with the LiFePO₄ and LiNi_{0.8}Co_{0.1}Mn_{0.1}O₂ cathodes to prepare full batteries with decent cycling performance. Overall, the Li₂S crystal orientation via sulfur defect engineering can serve as a general strategy to regulate dendrite-free Li metal deposition, and provide new insights to the development of high-performance Li metal batteries. The manuscript is well organized with a good logic flow, and the experiments have been rationally designed and conducted. Therefore, recommendation of this submission can be considered after the following comments and suggestions are addressed.

1. Both fast ion transport and efficient ion adsorption are important factors for uniform Li deposition. For the Li₂S(111)@Cu substrate, which is more important for Li deposition regulation?
2. The Cu clusters were generated during in-situ electrochemical lithiation of Cu₂S with the sulfur defects. What is the difference of facet between the Cu clusters and the native Cu substrate?
3. The authors claimed that Li₂S(111)@Cu could reduce the nucleation barrier, leading to more favorable Li deposition. I wonder if the Li₂S(111) still exists after the first cycle, and if the nucleation kinetics and Li deposition are still improved? This

should be examined and discussed in more details.

4. The solid-electrolyte interphase (SEI) chemistry derived from the Li₂S(111)@Cu, Li₂S@Cu, and bare Cu should be supplemented.

5. The influence of Li₂S crystal orientation on the Li transport dynamics was investigated by DFT calculations. More experimental evidences are suggested to supplement, e.g., the Li transference number and Li ion conductivity.

6. In Line 14, the authors claimed that the performance of full cell outperform the previously reported works, which is not fully accurate. The comparison should be better made by considering the N/P ratio, areal capacity, and current density. This requires careful revision in the revised manuscript.

7. In Figure 2b, it is difficult to see the detailed Coulombic efficiencies (CEs). All the plots containing the CE results should be rescaled or supplemented with an inset so that any fluctuations around 100% CE can be clearly observed. Further, for the determination of the electrochemical data (capacity, CE, cycling stability), multiple determinations should be performed and error bars should be provided to fully verify the differences between Li₂S(111)@Cu and bare Cu.

8. In Figures 2a and 3k, the schematic illustration of the highly orientated Li₂S arrays should re-draw according to the true morphology of the Li₂S arrays in Figure 3f. The current version will cause serious confusion and misunderstanding.

(Remarks on code availability)

Reviewer #3

(Remarks to the Author)

In this study, the authors engineered three-dimensional vertically aligned Li₂S(111)@Cu nanorod arrays as substrates for Li anodes. They manipulated the crystal orientation of Li₂S using sulfur-defect engineering to enhance Li⁺ transport and suppress dendritic growth. However, the study lacks depth in its exploration of material fabrication mechanisms and battery performance evaluations. I recommend substantial revisions before considering acceptance. Additionally, the authors should address the following points:

1. The authors prepared S-deficient Cu₂S@Cu NRs by annealing in an argon atmosphere, resulting in a small-angle shift in XRD characteristic peaks. They need to further explain the annealing-induced S-deficiency and the associated peak shifts.
2. In Figure 5c, for full battery testing, the battery using Cu as the anode substrate and LFP as the cathode showed slightly lower discharge capacity but better cycling stability compared to the Li₂S(111)@Cu-based battery. However, the abrupt termination after 200 cycles needs clarification.
3. While the authors demonstrated excellent cycle stability and high Coulombic efficiency in full cells, practical applications typically employ pouch cells. It is suggested that they extend their evaluations to high-load pouch cells to more convincingly demonstrate the performance and practical potential of the material.
4. To further validate the kinetic enhancements in lithium deposition and exfoliation provided by the Li₂S(111)@Cu anode substrate, it is advisable to conduct tests using symmetric or half-cell configurations at high current densities (3 or 5 mAh cm⁻²).
5. Considering that power batteries in practical applications often operate at 3 C or 5 C rates, the current tests with LiFePO₄ and NCM811 cathodes at only 0.5 C and 0.3 C are insufficient. The authors should consider conducting high-rate discharge tests to meet contemporary demands.

(Remarks on code availability)

Version 1:

Reviewer comments:

Reviewer #1

(Remarks to the Author)

The quality of the revised article has been improved, and it is recommended to accept it.

Reviewer #2

(Remarks to the Author)

In the revised manuscript, the authors have addressed my comments and suggestions, and the work has been significantly improved. I would like to recommend its publication in Nature Communications.

A minor error should be revised: in Supplementary Fig 65a, the unit of y axis should be mAh/g, rather than mAh/cm².

Reviewer #3

(Remarks to the Author)

The authors' efforts to include additional experiments are commendable. The revised manuscript has shown significant improvement, and I recommend its publication in Nature Communications after addressing the following issues:

1. Why was Li₂S chosen as the artificial SEI? Many studies have focused on inducing uniform lithium deposition through crystal surface engineering on Cu substrates. How does this work provide new insights or innovations compared to previous research?
2. I would appreciate a determination of the mass proportion of each component in the full cells, with particular emphasis on the artificial SEI, as this information is essential for assessing the commercial potential of this work.
3. I noted that the authors have added a cycling test for pouch cells. Can the morphology of the artificial SEI be maintained after cycling? It is recommended to include SEM and other relevant characterizations to confirm this.

Version 2:

Reviewer comments:

Reviewer #3

(Remarks to the Author)

The authors have addressed all my concerns, I recommend its publication in Nature Communications.

Sulfur defect engineering controls Li₂S crystal orientation for dendrite-free lithium metal batteries

Authors: Jin-Xia Lin[#], Peng Dai[#], Sheng-Nan Hu[#], Shi-Yuan Zhou, Gyeong-Su Park, Chen-Guang Shi, Jun-Fei Shen, Yu-Xiang Xie, Wei-Chen Zheng, Hui Chen, Shi-Shi Liu, Hua-Yu Huang, Ying Zhong, Jun-Tao Li, Xiaoyang (Jerry) Huang, Rena Oh^{*}, Wen-Feng Lin^{*}, Ling Huang^{*}, Shi-Gang Sun^{*}

We sincerely thank all the three reviewers for the constructive comments on our work. The valuable suggestions prompted us to engage in deeper reflection and further improve the quality of the manuscript. We have carefully revised the manuscript accordingly, and now provide a detailed point-by-point response to the reviewers' concerns below. The reviewer comments are laid out in *blue italicized font* and our responses are given in normal font for clarify. Additionally, all the changes in the revised manuscript and Supplementary Information have been highlighted **in yellow** for easy inspection.

Thank you again to all the reviewers for their insightful comments.

Responses to Reviewers:

To Reviewer #1 (Figs. R1-R15):	2-21
To Reviewer #2 (Figs. R16-R40):	22-43
To Reviewer #3 (Figs. R41-R47):	44-52
Reference	53-56

A Detailed Point-by-Point Responses to Reviewers' Comments

Reviewer #1: *The manuscript investigates the important of crystal orientation of Li₂S substrate for Li deposition behavior on developing high-performance lithium metal batteries. The author used DFT calculations to predict the close packed Li₂S(111) facet is favorable for achieving dense Li deposition. Two assemblies of the Li₂S(111)@Cu anode substrate with LiFePO₄ cathode that retained 88% capacity after 400 cycles, and with a commercial LiNi_{0.8}Co_{0.1}Mn_{0.1}O₂ cathode (4 mAh cm⁻²) that showcased 160 cycles. However, the manuscript is lack of innovations. Therefore, this manuscript is not recommended to be published in Nature Communication.*

Response: We would like to thank you for taking the time to review our manuscript and raising the concern regarding the innovations of our research. To enhance innovative impact of our work, we conducted an in-depth investigation into the mechanism of how S defects in Cu₂S_x@Cu can induce the crystal facet selectivity of Li₂S. This was achieved by observing intermediate phases and structural/compositional evolution in Cu₂S_x@Cu nanorods collected during the lithiation process, using advanced techniques such as HR-STEM/iDPC and HR-STEM-EELS, to effectively characterize light elements like S and Li atoms. The TEM results were included in **Fig. 2h** and **Supplementary Fig. 24-25** of the revised manuscript and Supplementary Information with corresponding analysis. Although understanding lithiation mechanism is crucial, studies based on the direct observations of lithiation intermediates remain scarce. While prior TEM observations of Cu₂S during lithiation were reported by Yi Cui group¹, this work uniquely observed lithiated Cu₂S_x with S defects, along with previously unidentified phases. These insights could play a significant role in advancing Li-dendrite free anodes.

We also want to emphasize again that while Li_2S material has been exemplified as a promising solid electrolyte interface (SEI) component that enables fast Li transport, the understanding regarding the influence of the Li_2S at Li/substrate interface and its surface structures on the Li nucleation and growth process are still absent, which plays a pivotal role in achieving dendrite-free Li deposition. Our research provides a comprehensive investigation into the impact of Li_2S at the Li/substrate interface and its exposed crystallographic facets on Li deposition, offering new insights into this area. We discovered the distinctive interactions between Li and the close-packed $\text{Li}_2\text{S}(111)$ serves as the most favorable facet for dendrite-free Li deposition, attributed to its effective adsorption and rapid transport of Li. This study introduces a novel approach by employing sulfur defect engineering to control Li_2S crystal orientation, achieving the construction of $\text{Li}_2\text{S}(111)@\text{Cu}$ material as Li anode substrate. More importantly, we identified a substrate-dependent Li nucleation process and a facet-dependent growth mode, revealed the profound impact of electrode substrate design and engineering on Li nucleation and growth behavior, that is essential for deployment of battery technologies. Therefore, we consider our work demonstrated innovation in the comprehensive analysis of theoretical models, the engineering design of materials, and the elucidation of structure-performance relationships.

We appreciate the reviewer for this input, the innovations of our work may not have been stated sufficiently before, we have improved our manuscript to clarify that, especially in the “Introduction” and “Discussion” sections. And thanks for your many valuable comments, we have carefully considered each suggestion and added extensive new experiments to address your concerns. We hope the revised manuscript is now suitable for publication in Nature Communications. Thank you for your time and consideration.

Revision: The corresponding contents have been added in the revised manuscript (page 10, line 15-23; page 11, line 1-11, line 12-14 and line 19-21; page 22, line 9-12).

Comment 1: What is the amount of sulfur defects in Cu_2S_x ? How to regulate the content of sulfur defects. How does the sulfur defect content regulate the Li_2S crystal plane and how does it affect the Li deposition process?

Response to Comment 1: Thank you for your valuable comment, we have conducted further discussions and experiments to address your questions. Specifically, we synthesized Cu_2S -200 with a different S defect content from that of Cu_2S_x , and thoroughly investigated its Li activation process, focusing on the formation of the corresponding Li_2S -based substrate. Whereafter, we illustrated the effects of different Li_2S -based substrates on Li deposition behavior and battery performance. Following is the detailed point-by-point responses.

Calculation of the content of sulfur defects: Based on the XPS (**Supplementary Fig.6**) and EDS results (**Supplementary Fig.13**), we could obtain the amount of S defects in Cu_2S_x is 12.7 atomic% from XPS and 14.5 atomic% from EDS. The calculation is conducted based on the atomic ratio of Cu:S of Cu_2S_x and Cu_2S .

Regulation of the content of S defects: To regulate the content of S defects, the annealing temperature is a crucial factor. The effect of annealing temperature tremendously influences the migration of S atoms and the thermodynamic driving force^{2,3}. The annealing temperature of Cu_2S_x is 280°C, when we tuned the temperature to 200°C and kept other conditions constant, Cu_2S -200 materials could be obtained. The electron paramagnetic resonances (EPR) measurement was employed to investigate the variation of S

defects content. As shown in **Fig. R1**, the peak intensity of Cu₂S-200 decreases when compared with Cu₂S_x, indicating a decrease of S defects concentration⁴. To determine to S defect content in Cu₂S-200, EDS and XPS measurement were further conducted (**Fig. R2**). Based on the Cu:S atomic ratio data, we obtained the amount of defect sulfurs is 7.1 atomic% and 8.6 atomic% from XPS and EDS, respectively. These results collectively evidenced that the annealing temperature is crucial for regulating the S defect content.

Fig. R1| EPR spectra of Cu₂S-200 compared to Cu₂S_x. The peak intensity is proportional to the concentration of S defects.

Fig. R2| The atomic ratios of Cu:S from the XPS survey spectra and EDS data for Cu₂S-200 material.

Different S defect content in regulating the Li_2S crystal plane: During the reaction between Cu_2S and Li, the S sublattice rearranged as Cu left the structure, and the close-packed S planes shifting from the hcp-type stacking of Cu_2S to the fcc-type stacking of Li_2S ¹. In the high chalcocite Cu_2S , the crystallography of the structure is defined by the hexagonal sulfur framework (space group $Fm\bar{3}m$)⁵. Various in S defect content may alter the arrangement of S sublattice, thus potentially influencing the crystal orientation of Li_2S . To illustrate the effect of the S defect content on regulating Li_2S crystal planes, in-situ XRD characterization was performed to monitor the electrochemical lithiation of $\text{Cu}_2\text{S-200@Cu}$. As shown in **Fig. R3**, $\text{Cu}_2\text{S-200@Cu}$ exhibited similar behaviors like $\text{Cu}_2\text{S@Cu}$ and $\text{Cu}_2\text{S}_x\text{@Cu}$ during the initial Li activation process, such as the peak shifts and the formation of intermediate phase. However, unlike the lithiation product of Cu_2S_x , the Li_2S phase formed from $\text{Cu}_2\text{S-200@Cu}$ showed less crystal surface selectivity towards (111) facet with additional weak reflections of (220) and (311) facets. The eventual material after Li activation of $\text{Cu}_2\text{S-200@Cu}$ was denoted as $\text{Li}_2\text{S-200@Cu}$. In detail, the intensity ratios of (111), (220), (311), and (222) reflections from cubic Li_2S in $\text{Li}_2\text{S@Cu}$ and $\text{Li}_2\text{S-200@Cu}$ are 1:1.31:0.70:1.74 and 1: 0.28:0.23:0.25, respectively. These results suggest that compared to $\text{Cu}_2\text{S}_x\text{@Cu}$, $\text{Cu}_2\text{S-200@Cu}$ with lower S defects could not construct $\text{Li}_2\text{S}(111)\text{@Cu}$, however, relative to $\text{Li}_2\text{S@Cu}$, $\text{Li}_2\text{S-200@Cu}$ exhibited an increased propensity to exhibit crystallographic facet selectivity.

Fig. R3 | In situ XRD 1D maps and corresponding discharge curve of $\text{Li}_2\text{S-200@Cu}$ formation process (the dashed lines are the positions where the peaks shift occurs).

Moreover, to explore the mechanism of how S defects in Cu_2S_x can induce the crystal selectivity in Li_2S , $\text{Cu}_2\text{S}_x\text{@Cu}$ NRs were collected in the middle of lithiation process toward forming $\text{Li}_2\text{S(111)@Cu}$ and analyzed by TEM. As demonstrated in **Fig. R4**, during the lithiation process, Cu- and S-deficient regions were observed in the Li-intercalated Cu_2S_x region, Li_yCuS_x . In and above this Li-intercalated region, we observed crystalline Li_xCu and amorphous Li_xS as polycrystalline nanoparticles. The verification of the presence of the aforementioned intermediates by HAADF-STEM/iDPC and STEM-EELS can be seen in **Figs. R5-R6**. These observations, along with the in situ XRD result shown in **Supplementary Fig. 16** in the revised manuscript, suggest that S defects-driven Li intercalation might facilitate the diffusion of Cu and S from Cu_2S_x region, forming Li_xCu , Cu, and potentially Li_xS species. As lithiation is further

processed, we hypothesis that the amorphous Li_xS nanoparticles may transform into the cubic Li_2S nanoparticles predominantly exposing the thermodynamically most stable (111) facet⁶.

Fig. R4| The structural and compositional evolution occurring during the lithiation.

Fig. R5| TEM characterizations of $\text{Cu}_2\text{S}_x@Cu$ collected during lithiation. (a) HR-TEM image showing stacking faults in Li intercalated Cu_2S_x shell. The arrows in a1 and a2 region indicate the presence of a stacking fault. (b) High-resolution HAADF/iDPC-STEM images showing Cu-deficient regions in Cu_2S_x shell with Cu voids. The red solid lines and arrows highlight the Cu-deficient regions. (b: HAADF-STEM, b1 and b2: iDPC-STEM image). (c) HAADF-STEM image and (d,e) STEM-EELS spectra indicating (c1) a Cu-deficient with darker contrast and (c2) S-deficient region with brighter contrast.

Fig. R6| TEM characterizations of $\text{Cu}_2\text{S}_x@Cu$ collected during lithiation. (a) TEM/HR-TEM images with the corresponding FFT diffractogram showing (a1) Cu_2S core, (a2) Li intercalated Cu_2S_x shell, (a3) nanocrystallites and amorphous phase (a3). (b,c) High resolution HAADF/iDPC-STEM image of nanocrystallites and amorphous phase on Li_yCuS_x . (d,e) HR-TEM images with the corresponding FFT diffractogram of (d) Li_xCu and (e) hcp LiCu . (f,g) HAADF-STEM images and (h) STEM-EELS spectra

showing distinguished Li, Cu, and S concentrations in f1, f2, g1, and g2 region marked in f and g.

Effects on Li deposition behavior: To illustrate how the S defects affect the Li deposition process, different amounts of Li were deposited on Li₂S-200@Cu substrate. As SEM images shown in **Fig. R7**, Cu₂S-200@Cu exhibited 3D vertical nanorods (NRs) array as Cu₂S@Cu (**Supplementary Fig.10a-10d**), while with a reduced density. After Li activation, the morphology of Li₂S-200@Cu substrate was present in **Figs. R8a** and **R8d**. During the initial stage of Li deposition, such as deposited 0.001 and 0.002 mAh cm⁻² Li (**Figs. R8b** and **R8e**, **Figs. R8c** and **R8f**), two manners of deposition were observed on the Li₂S-200@Cu substrate's surface. Specifically, some of Li particles adhered on the Li₂S-200@Cu NRs surface in a surface-contact manner, as marked in gold dashed circles in **Fig. R8e-R8f**, this mode is similar with that of manner observed on the Li₂S(111)@Cu NRs surface (**Supplementary Fig. 28**). Meanwhile, part of Li particles displayed a point-contact manner (highlighted by red dashed circles), as that of mode observed on the Li₂S@Cu NRs surface (**Supplementary Fig. 30**). When the deposition amount increased to 1 mAh cm⁻², the aggregated Li displayed a seaweed-like clusters morphology, distributed in the Li₂S-200@Cu NRs array, showing a shaggy structure (**Fig. R9a**). This is different from the completely enveloped formed in the Li₂S(111)@Cu NRs by deposited Li (**Fig. 4h**). As the Li deposition amount increase, SEM image as shown in **Fig. R9b** reveal a partially dense Li layer on the top surface Li₂S-200@Cu NRs substrate. Noteworthy, the Li₂S-200@Cu NRs, Li₂S@Cu NRs and Li₂S(111)@Cu NRs have similar 3D vertical structure, the various Li deposition behavior primary attributed to differences in Li₂S crystal facets. Therefore, these experimental results suggest that Li deposited on various Li₂S-based substrates became smoother with an increasing proportion of the (111) facet, consisting with our previous findings that the Li₂S(111) plane, with high Li affinity and low diffusion barrier plays a crucial role for the lateral growth

of Li.

Fig. R7| SEM images of $\text{Cu}_2\text{S-200@Cu}$.

Fig. R8| SEM images of Li deposited on $\text{Li}_2\text{S-200@Cu}$ substrate. (a, d) Initial $\text{Li}_2\text{S-200@Cu}$ substrate. (b, e) After depositing $0.001 \text{ mAh cm}^{-2}$ and (c, f) $0.002 \text{ mAh cm}^{-2}$ Li.

Fig. R9 | SEM images of Li deposited on Li₂S-200@Cu substrate. (a) 1 mAh cm⁻² Li and (b) 10 mAh cm⁻² Li.

Effects on electrochemical performance in battery system: So far, we have demonstrated that S defects play a crucial role in achieving crystallographic facet selectivity of Li₂S-base substrate, the regular crystal orientation of Li₂S(111) facilitate the smooth Li deposition morphology. To investigate the influence of Li₂S crystal plane on Li deposition kinetics, linear sweep voltammetry (LSV) measurements were conducted. The results, as shown in **Fig. R10**, reveal a gradual increase in Li plating kinetics as the exposed (111) plane of Li₂S increase. To further assess the effects of Li₂S crystal plane in battery systems, the Coulombic efficiency (CE) test program follows a partial-stripping mode was measurement. As shown in **Fig.R11**, the average CE based on Li₂S-200@Cu substrate could reach 99.1% in at a current density of 1 mA cm⁻² with an areal capacity of 1 mAh cm⁻². When compared with Li₂S@Cu and Li₂S(111)@Cu substrates under the same conditions, the value of CE display an order of Li₂S(111)@Cu > Li₂S-200@Cu > Li₂S@Cu (**Fig. 4a** and **Supplementary Fig. 45**). The inset of **Fig.R11** showed the SEM image of Li₂S-200@Cu substrate after Li completely stripped, we observed that the morphology of NRs array was largely preserved while a small amount of inactive Li residue present. To further evaluate the reversibility of Li deposition/dissolution, long-term cycling test was carried out under a plating-full-stripping mode. As

shown in **Fig.R12a**, $\text{Li}_2\text{S-200@Cu}$ exhibited stable cycling around 300 cycles, such a lifespan falling between the $\text{Li}_2\text{S@Cu}$ (250 cycles) and $\text{Li}_2\text{S(111)@Cu}$ (500 cycles) substrates. The inset of **Fig. R12a** displayed the morphology of three Li_2S -based substrates after 50 cycles, the structural integrity of substrates and the residual inactive Li, align well with the trend of cycle life. Hitherto, $\text{Li}_2\text{S-200@Cu}$ demonstrated faster Li plating kinetics, higher Li utilization, and improved Li plating/stripping reversibility compared to the $\text{Li}_2\text{S@Cu}$ substrate, while it remained inferior to the $\text{Li}_2\text{S(111)@Cu}$ substrate. The effects of Li_2S crystal plane in practical system was assessed by constructing full cells using commercial $\text{LiNi}_{0.8}\text{Co}_{0.1}\text{Mn}_{0.1}\text{O}_2$ (NMC811) with a high areal loading of 20 mg cm^{-2} . As shown in **Fig. R12b**, the full cell based on $\text{Li}_2\text{S-200@Cu}$ anode substrate showcased a lifespan of 107 cycles with 80% capacity retention and maintained average CEs of 99.3%, lying between the 86 cycles of $\text{Li}_2\text{S@Cu}$ and 160 cycles of $\text{Li}_2\text{S(111)@Cu}$. **Fig.R12c-R12e** presented the corresponding charge/discharge profiles of three Li_2S -based substrates, the limited voltage drop in $\text{Li}_2\text{S-200@Cu}$ system indicated an enhancement in interface stability, which could be attributed to the optimized Li plating/stripping process. In summary, the battery performances based on $\text{Li}_2\text{S-200@Cu}$ substrate exhibited optimizations compared to that of $\text{Li}_2\text{S@Cu}$ substrate, revealing the role of $\text{Li}_2\text{S(111)}$ plane in promoting smooth Li deposition. Meanwhile, the deteriorated performance compared to the $\text{Li}_2\text{S(111)@Cu}$ substrate, further highlighting the vital role of the homogenous $\text{Li}_2\text{S(111)}$ plane in maintaining dendrite-free Li deposition during cycling process, aligning well with our previous results.

Fig. R10 | LSV curves of various Li||substrate half cells.

Fig. R11 | CE test of half cells utilizing $\text{Li}_2\text{S-200@Cu}$ substrate, with the inset showing a SEM image of the substrate after complete Li stripping.

Fig. R12 | Electrochemical performance of Li||substrate half cells and NCM811 full cells based on

Li₂S@Cu, Li₂S-200@Cu and Li₂S(111)@Cu substrates. (a) CE versus cycle number under a plating-full-stripping mode at 0.5 mA cm⁻² with 0.5 mAh cm⁻², the insets are SEM images of the substrates after 50 cycles. (b) Discharge capacity retention of Li-substrate||NCM811 full cells at 0.3 C rate. (c-e) Corresponding GDC profiles of NCM811 full cells at different cycles based on Li₂S@Cu (c), Li₂S-200@Cu (d) and Li₂S(111)@Cu (e) substrates. (To visually show the variation in electrochemical performance of various Li₂S-based substrates with cycling, and better reveal the effect of Li₂S crystal plane on battery performance, we extracted the data of Li₂S@Cu (from Supplementary Fig. 47 and Supplementary Fig. 67) and Li₂S(111)@Cu (from Fig. 4b and Fig. 5e-5f), and plotted them along with the Li₂S-200@Cu substrate data.)

Revision to Comment 1: The corresponding contents have been added in the revised manuscript (page 7, line 5-7; page 8, line 15-18; page 10, line 20-23; page 11, line 1-23; page 12, line 1-14 and 19-21; page 13, line 5-8; page 17, line 13-17; page 20, line 8-10) and the Supplementary Figs. 20-22, 24-25, 36-37, 54-55 and 66, and the Supplementary Note 1 of defect effects.

Comment 2: How to ensure the morphology and uniformity of Li₂S deposition during the conversion process between Cu₂S_x and Li₂S?

Response to Comment 2: Thank you for raising this issue. Based on the SEM images (**Supplementary Fig. 11, Supplementary Fig. 28a, 28d**), we did observe that when Cu₂S_x was converted to Li₂S, there were no significant changes in the morphology and array structure, which is additionally corroborated by **Fig. R13**. In our TEM data (**Supplementary Fig. 12a and Fig. 2f**), we noticed that the microstructure of Cu₂S_x and Li₂S nanorods was different, the nanorod consisted of many stacked nanoparticles after the conversion process. It should be noted that Cui et al. have utilized in situ TEM to study structural and morphological transformation of individual Cu₂S nanocrystal during reaction with Li¹. They observed that the Cu₂S crystal transformed to the Li₂S phase and maintained the original shape, revealing the key to the negligible change in particle morphology was the structural similarities between the two phases. Following

the TEM observations (**Fig. 2h** and **Supplementary Figs. 24-25**) and in situ XRD data (**Fig. 2e** and **Supplementary Fig. 19**), we could suggest that $\text{Cu}^{+\delta}$ of Cu_2S_x or Cu_2S nanorods (JCPDS No. 26-1116, space group $P6_3/mmc$) were replaced by Li^+ to generate Li_2S (JCPDS No.26-1188, space group $Fm\bar{3}m$). A nanorod may be composed of several crystal grains and each grain may transform to Li_2S grain and Cu, which made the surface rougher with Li_2S particles than the Cu_2S or Cu_2S_x nanorod surface. And the similar S sublattice stacking of ABAB in Cu_2S to the ABCABC in Li_2S attributing the generated Li_2S almost retained the morphology of Cu_2S or Cu_2S_x . Regarding the uniformity of Li_2S deposition, we would like to emphasize that the synthesized $\text{Li}_2\text{S}@Cu$ or $\text{Li}_2\text{S}(111)@Cu$ substrate is not a layer of Li_2S deposited on Cu substrate. There have been reported that the reaction between Cu_2S and Li was more likely to a displacement reaction, the newly formed Cu domains attached to the surface of Li_2S ^{1,8}. In **Fig. R14**, we carried out elemental mapping of $\text{Li}_2\text{S}@Cu$ nanorod, illustrating a homogeneous distribution of Cu and S. These results collectively evidenced the Cu_2S or Cu_2S_x nanorod transformed to the mixture of Li_2S nanoparticle and Cu nanoparticle after the lithiation process.

Fig. R13| TEM images of the morphology evolution occurring during the lithiation of Cu_2S core- Cu_2S_x shell NR. (a) Before lithiation. (b) During lithiation. (c) After cycling.

Fig. R14| High-magnification elemental mapping of $\text{Li}_2\text{S}(111)\text{@Cu}$ nanorod.

Comment 3: Generally, Li_2S has poor electronic conductivity. After depositing a layer of Li_2S on the Cu current collector, how can it promote the occurrence of Li deposition electrochemical reactions? How does Li_2S regulate the deposition behavior of lithium when lithium deposition covers the Li_2S layer?

Response to Comment 3: Thank you for your comment. Following is our point-by-point responses about your two questions:

(1) Li_2S has poor electronic conductivity. After depositing a layer of Li_2S on the Cu current collector, how can it promote the occurrence of Li deposition electrochemical reactions?

We agree with the reviewer's viewpoint that Li_2S typically exhibits low electronic conductivity, however, we would like to emphasize again that Cu_2S or Cu_2S_x nanorod transformed to the mixture of Li_2S nanoparticle and Cu nanoparticle after the lithiation process. Thus, the high ionic conductivity of Li_2S together with high electronic conductivity of Cu facilitates the charge transfer during the Li deposition process. As we described in the manuscript, these arrays served as a mixed ion/electron conductor, effectively balanced both electron and ion transport at the Li/substrate interface (page 3, line 15-16).

(2) How does Li_2S regulate the deposition behavior of lithium when lithium deposition covers the Li_2S layer?

The Li₂S mainly impact the initial deposition behavior of Li, including the initial nucleation and the following growth stages. As the DFT results in **Fig. 1**, we discovered the crystal orientation-dependent nature of Li nucleation and growth on Li₂S surfaces. Compared to the Li₂S(311) and Li₂S(110), the Li₂S(111) plane exhibited the highest Li affinity and the lowest diffusion barrier, which could robust Li adsorption in the Li nucleation process and facilitate rapid Li transport in the Li growth process. The **Fig. 3f-3g** and **Supplementary Fig. 30** showed the SEM images of trace amount of Li (0.001 and 0.002 mAh cm⁻²) deposited on Li₂S(111)@Cu and Li₂S@Cu substrate, respectively. The Li particles were homogeneously distributed on Li₂S(111)@Cu surface with a surface-contact manner, while minute Li adhered on Li₂S@Cu surface in a point-contact. As the deposition amount increased, **Fig. 3h-3j** revealed the deposited Li could completely covered the Li₂S(111)@Cu nanorods (1 mAh cm⁻²), and formed a smooth Li deposition layer (1 mAh cm⁻²). However, a loosely Li deposited layer was formed on the top surface of Li₂S@Cu NRs substrate (**Supplementary Figs. 31**). In summary, the Li₂S could regulate the Li deposition by tuning surface chemistry to vary the abilities of bind and transport Li, modifying the initial Li nucleation and growth behavior, which strongly affected the texture of deposition layer after the deposited Li covered the Li₂S.

***Comment 4:** Is there a basis for the reaction mechanism of sulfur vacancy regulation of electronic structure changes in Page 10? The author should elaborate the changes in the mechanism in detail.*

Response to Comment 4: We appreciate the reviewer for pointing out the missing elaboration of the mechanism of S defect regulation of electronic structure changes. It has been reported that defect engineering is a crucial strategy to modulate the surface electronic structure of materials⁹. The presence

of defects could disrupt the periodic crystalline structure of materials, and then alter the surface electronic structure with localized electron re-distribution¹⁰. Specifically, after inducing S defects in Cu₂S@Cu, the electron density around Cu atoms increased, as indicated by the lower Cu binding energies and higher S binding energies in XPS spectra of Cu₂S_x@Cu (**Supplementary Fig. 5**). The increased EPR signal reflected more unpaired electron in Cu₂S_x than Cu₂S (**Fig. 2b**). The XAS results revealed a lower valence state of Cu and a decreased coordination number in Cu₂S_x than in Cu₂S (**Supplementary Figs. 7-8**). All these results collectively evidenced the introduction of S defect could change the electronic structure of the material. In addition, as we mentioned in the **Supplementary Note 1** of defect effect, the introduction of defects can regulate the conductivity of materials, influencing the density and activity of active sites¹. In this work, the sulfur defect of Cu₂S_x (20.9 Ω) led to a higher conductivity compared to Cu₂S (51.3 Ω), which directly impacts the electron-transfer rate during the Li activation process. Notably, the intermediate phase in the Cu₂S_x system appeared earlier and persisted longer than in the Cu₂S system (**Supplementary Figs. 15 and 18**). The surface of Cu₂S_x NRs exhibited increased roughness than Cu₂S NRs, as observed in the TEM images in **Supplementary Figs. 10-11**, indicating an increased number of active sites during lithiation process. The online differential electrochemical mass spectrometry (OEMS) tests (**Supplementary Fig. 26**) detected the C₂H₄ and CO₂ gases during the formation of Li₂S@Cu, which originated from the decomposition of electrolyte¹¹, indicating electrons transfer from the Cu₂S@Cu to the lowest unoccupied molecular orbital (LUMO) of the electrolyte. However, no gas evolution occurred as the Li₂S(111)@Cu was forming, indicating a distinct electron transfer mechanism from the electrode to the electrolyte compared to the Cu₂S system. These phenomena may be attributed to S defect regulating the electronic structure, leading to a lower Fermi level of Cu₂S_x than Cu₂S, thus suppressing the electron

transfer to the electrolyte. In conclusion, the introduction of defect engineering altered the sublattice arrangement of Cu_2S_x , regulating the electronic structure of material, which has a significant impact on the crystal orientation of Li_2S as we discussed in detail in the **Response to Comment 1** (*Different S defect content in regulating the Li_2S crystal plane*).

Revision to Comment 4: The corresponding discussions have been added in the revised manuscript (page 10, line 1-23; page 11, line 1-14 and 19-21) and Supplementary Note 1 of defect effects.

Comment 5: Is a sudden drop in voltage in Fig.5b a short circuit? The author needs to provide relevant data.

Response to Comment 5: We appreciate the reviewer's comment regarding the sudden drop in voltage in Fig. 5b. We feel sorry to clarify that there was a missing of marking the current density, the sudden drop in voltage was due to the test current density abruptly changed from 4 mA cm^{-2} to 0.5 mA cm^{-2} . The test was carried out by gradually increasing the current density from 0.5, 1, 2, 3, to 4 mA cm^{-2} with a fix capacity of 1 mAh cm^{-2} to evaluate the rate capability. After that, we returned the current density from 4 back to 0.5 mA cm^{-2} to further assess the reversibility and long-term cycling stability of batteries. We have added annotations to **Fig. 5b**, as displayed in **Fig. R15**, and made a clearer description in the revised manuscript.

Fig. R15| Electrochemical performance of the symmetric cells based on Cu and Li₂S(111)@Cu substrates. Rate performance at different current densities from 0.5 to 4 mA cm⁻² with an areal capacity of 1 mAh cm⁻², and following long-term cycling test at 0.5 mA cm⁻² with 1 mAh cm⁻².

Revision to Comment 5: The corresponding Figure has been updated in Fig. 5b, and the corresponding discussions have been added in the revised manuscript (page 18, line 21-22; page 19, line 1).

Reviewer #2: *The submission by Sun et al. reported a three-dimensional $\text{Li}_2\text{S}(111)@\text{Cu}$ nanorod array as the Li deposition substrate through sulfur defect engineering for Li_2S crystal orientation control, which exhibited efficient regulation of Li-ion adsorption, nucleation, and deposition. The authors investigated the interactions between Li and various surface structures of Li_2S , and revealed that $\text{Li}_2\text{S}(111)$ plane exhibits the highest Li affinity and lowest diffusion barrier, resulting in dense Li deposition. The $\text{Li}_2\text{S}(111)@\text{Cu}$ substrate showed a mixed Li nucleation mode during initial deposition followed by a mono-instantaneous growth mode, which facilitated smooth Li deposition. The $\text{Li}_2\text{S}(111)@\text{Cu}$ anode substrates can pair with the LiFePO_4 and $\text{LiNi}_{0.8}\text{Co}_{0.1}\text{Mn}_{0.1}\text{O}_2$ cathodes to prepare full batteries with decent cycling performance. Overall, the Li_2S crystal orientation via sulfur defect engineering can serve as a general strategy to regulate dendrite-free Li metal deposition, and provide new insights to the development of high-performance Li metal batteries. The manuscript is well organized with a good logic flow, and the experiments have been rationally designed and conducted. Therefore, recommendation of this submission can be considered after the following comments and suggestions are addressed.*

Response: We sincerely appreciate your recognition and general summary of our results, we have seriously taken your valuable comments and suggestions into consideration, and provided a detailed point-by-point response as follows.

Comment 1: *Both fast ion transport and efficient ion adsorption are important factors for uniform Li deposition. For the $\text{Li}_2\text{S}(111)@\text{Cu}$ substrate, which is more important for Li deposition regulation?*

Response to Comment 1: We are thankful to the comment. Since the high ionic conductivity of Li_2S

could guarantee the fast ion transport, we suggested that, in comparison to the $\text{Li}_2\text{S}@Cu$ substrate, the effective ion adsorption of $\text{Li}_2\text{S}(111)@Cu$ substrate to provide uniform lithophilic nucleation sites was more important for regulating Li deposition. To verify this hypothesis, we measured the Li transference number (t_+) based on different substrates, as expected, the t_+ values of the $\text{Li}_2\text{S}(111)@Cu$ ($t_+ = 0.68$) and $\text{Li}_2\text{S}@Cu$ substrate ($t_+ = 0.64$) showed no significant differences. Detailed discussion of t_+ will be presented in **Response to Comment 5**. However, it should be noted that the dense deposition of Li on the $\text{Li}_2\text{S}(111)@Cu$ substrate was attributed to the synergistic effects of the effective adsorption and rapid transport of Li, both of them were crucial and indispensable.

Revision to Comment 1: The corresponding contents have been added in the Supplementary Fig. 60.

Comment 2: The Cu clusters were generated during in-situ electrochemical lithiation of Cu_2S with the sulfur defects. What is the difference of facet between the Cu clusters and the native Cu substrate?

Response to Comment 2: Thank you for raising this issue. To address your concerns, we summarized the peak intensity variations of the detected signals of Cu(111) and Cu(200) during the in-situ electrochemical lithiation of $\text{Cu}_2\text{S}@Cu$ and $\text{Cu}_2\text{S}_x@Cu$. As shown in **Fig. R16a-R16b**, both of the intensities of Cu(111) and Cu(200) underwent an initial increase, followed by stabilization as the lithiation progresses. Meanwhile, the intensity of Cu(111) remained higher than Cu(200) throughout, suggesting the Cu clusters in $\text{Li}_2\text{S}@Cu$ and $\text{Li}_2\text{S}(111)@Cu$ substrates exhibited the (111) facet dominated as the native Cu. **Fig. R16c** displayed the variations of the peak intensity ratio of Cu(111)/Cu(200), the value in $\text{Cu}_2\text{S}@Cu$ and $\text{Cu}_2\text{S}_x@Cu$ were 1.67 and 1.81, respectively; after Li activation process, the value in $\text{Li}_2\text{S}@Cu$ and

$\text{Li}_2\text{S}(111)@\text{Cu}$ increased to 2.08 and 2.31, respectively. These changes suggested that the increment of the proportion of (111) facets in the Cu clusters. Overall, the Cu clusters formed through the in-situ electrochemical lithiation of Cu_2S and Cu_2S_x both exhibited the (111) facet dominated, as well as the native Cu substrate. Therefore, the variations in battery performance were mainly attributed to the different crystal facets of the generated Li_2S .

Fig. R16| The characteristic peak intensity variations of Cu(111) and Cu(200) during the in-situ electrochemical lithiation process of $\text{Cu}_2\text{S}@\text{Cu}$ and $\text{Cu}_2\text{S}_x@\text{Cu}$. (a) The peak intensity variations during the formation process of $\text{Li}_2\text{S}@\text{Cu}$ substrate. (b) The peak intensity variations during the formation process of $\text{Li}_2\text{S}(111)@\text{Cu}$ substrate. (c) The peak intensity ratio of the Cu(111)/Cu(200).

Revision to Comment 2: The corresponding contents have been added in the revised manuscript (page 10, line 10-12) and the Supplementary Fig. 23.

Comment 3: The authors claimed that $\text{Li}_2\text{S}(111)@\text{Cu}$ could reduce the nucleation barrier, leading to more favorable Li deposition. I wonder if the $\text{Li}_2\text{S}(111)$ still exists after the first cycle, and if the nucleation kinetics and Li deposition are still improved? This should be examined and discussed in more details.

Response to Comment 3: We would like to express our sincere appreciation for the valuable comment and suggestion. We have conducted further experiments and discussion to better address your concerns.

As displayed in **Fig. R17a**, the XRD pattern of $\text{Li}_2\text{S}(111)@\text{Cu}$ substrate after the first cycle indicated the existence of $\text{Li}_2\text{S}(111)$. The LSV curves in **Fig. R17b** demonstrated that the cycled $\text{Li}_2\text{S}(111)@\text{Cu}$ substrate continued to enhance the Li plating kinetics. To further examined the substrate effects in regulating Li deposition after the first cycle, we plated different amounts of Li on three cycled substrates and shown the SEM results in **Figs. R18-R21**. The 3D vertical array structure of the $\text{Li}_2\text{S}(111)@\text{Cu}$ substrate remained and almost no inactive Li could be observed after the first cycle (**Fig. R18a, R18d**). After plating a trace amount of Li (0.001 and $0.002 \text{ mAh cm}^{-2}$) on the cycled $\text{Li}_2\text{S}(111)@\text{Cu}$ substrate, the sphere-shaped Li particles appeared (**Fig. R18b, R18e**), and then aggregated (**Fig. R18c, R18f**) as the behavior on the fresh $\text{Li}_2\text{S}(111)@\text{Cu}$ substrate. Regarding to the $\text{Li}_2\text{S}@\text{Cu}$ and Cu substrates, where the SEM images displayed limited residual inactive Li after the first cycle (**Fig. R19a and Fig. R20a**). After plating 0.001 and $0.002 \text{ mAh cm}^{-2}$ Li, the Li particles were adhered on the cycled $\text{Li}_2\text{S}@\text{Cu}$ substrate in a point-contact configuration (**Fig. R19b, R19e**). On the cycled Cu substrate, the irregularly deposited Li particles quickly aggregated to form dendrites (**Fig. R20b-R20c**). As the plating Li amount increased to 1 and 10 mAh cm^{-2} , the deposited Li could entirely cover the cycled $\text{Li}_2\text{S}(111)@\text{Cu}$ nanorods and eventually form a dense Li layer (**Fig. R21a, R21d**). Whilst with the cycled $\text{Li}_2\text{S}@\text{Cu}$ substrate, the plated Li did not fully envelop the nanorods (**Fig. R21b**) due to the lack of regularity in the crystal orientation, resulting in a loose Li deposited layer (**Fig. R21e**). SEM images of the cycled Cu substrate after plating 1 and 10 mAh cm^{-2} Li were provided in **Fig. R21c and R21f**, which exhibited substantially Li dendrites on the surface, driving from the poor Li affinity and large diffusion barrier. Taking all these results into consideration, the morphological and crystal structure of $\text{Li}_2\text{S}(111)@\text{Cu}$ could be well maintained after the first cycle, continuing to enhance Li deposition kinetics and uniform Li deposition.

Fig. R17| Characterizations of various substrates after the first cycle. (a) XRD patterns of $\text{Li}_2\text{S}@Cu$ and $\text{Li}_2\text{S}(111)@Cu$ substrates after the first cycle. (b) LSV curves obtained from $\text{Li}||\text{substrate}$ half cells after the first cycle.

Fig. R18| SEM images of Li deposited on $\text{Li}_2\text{S}(111)@Cu$ substrate after the first cycle. (a, d) $\text{Li}_2\text{S}(111)@Cu$ substrate after the first cycle. (b, e) After deposited $0.001 \text{ mAh cm}^{-2}$ Li. (c, f) After deposited $0.002 \text{ mAh cm}^{-2}$ Li.

Fig. R19 | SEM images of Li deposited on $\text{Li}_2\text{S}@\text{Cu}$ substrate after the first cycle. (a, d) $\text{Li}_2\text{S}@\text{Cu}$ substrate after the first cycle. (b, e) After deposited $0.001 \text{ mAh cm}^{-2}$ Li. (c, f) After deposited $0.002 \text{ mAh cm}^{-2}$ Li.

Fig. R20 | SEM images of Li deposited on Cu substrate after the first cycle. (a) Cu substrate after the first cycle. (b-c) After deposited b) $0.001 \text{ mAh cm}^{-2}$ and c) $0.002 \text{ mAh cm}^{-2}$ Li.

Fig. R21 | SEM images of Li deposited on different substrates after the first cycle with 1 and 10 mAh cm⁻². (a, d) Li₂S(111)@Cu substrate. (b, e) Li₂S@Cu substrate. (c, f) Cu substrate.

Revision to Comment 3: The corresponding contents have been added in revised manuscript (page 13, line 2-5) and Supplementary Figs. 32-35.

Comment 4: The solid-electrolyte interphase (SEI) chemistry derived from the Li₂S(111)@Cu, Li₂S@Cu, and bare Cu should be supplemented.

Response to Comment 4: Thank you for this important suggestion. We have conducted XPS depth profiling measurement to analyze the chemistry information of the SEI formed on the three substrates after 50 cycles. In the C 1s spectra (**Fig. R22**), the peaks at 286.4 and 288.5 eV were assigned to C-OR and C=O from the DOL/DME residue or the decomposition products, respectively¹². The peaks at 289.9 and 292.8 eV corresponded to CO₃²⁻ and C-F, which were attributed to the decomposition of solvent and

LiTFSI residue¹³, respectively. For the F 1s spectra, two distinct peaks at 684.9 and 688.6 eV were ascribed to LiF and C-F¹⁴, respectively (**Fig. R23**). It should be noted that the high electron tunneling barrier of LiF could enhance the interface stability¹⁵. As the sputtering depth increased, the LiF dominated in the F 1s spectra for all samples, especially in the Li₂S(111)@Cu system, suggesting the less solvent decomposition. In S 2p spectra (**Fig. R24**), the peaks at 161-165 eV and 166-170 eV were corresponded to the Li₂S/Li₂S₂ and SO₄²⁻/SO₃²⁻ species¹⁶. Noteworthy, the contents of the sulfide species from all the samples were similar, indicating that these sulfide signals generated from the SEI on the Cu, Li₂S@Cu and Li₂S(111)@Cu substrates. For a clear evaluation of the SEI chemistry structure, we summarized the variation of atomic ratios with the sputtering depth in **Fig. R25** and **Table R1**. The higher C atomic ratios and lower F atomic ratios in the SEI formed on the Cu and Li₂S@Cu substrates (**Fig. R25a-R25b**), indicate the solvent decomposition. The C content quickly decreased after sputtering, indicating the organic species resulting from the solvents decreased in the inner layer. **Fig. R25c** shows that the SEI formed on the Li₂S(111)@Cu substrate has the lowest C and highest F atomic percentages, indicating a minimal presence of organic and a predominant inorganic components. In addition, there was no obvious difference in the S atomic ratios among all samples, verifying that the improved performance of batteries could not be ascribed to the Li₂S SEI in our studies. The specific effects of the substrate to the SEI chemistry will be further investigated in follow up studies.

Fig. R22| The normalized XPS fitting results for C 1s of different substrates after 50 cycles. (a) Cu. (b) $\text{Li}_2\text{S}@\text{Cu}$. (c) $\text{Li}_2\text{S}(111)@\text{Cu}$.

Fig. R23| The normalized XPS fitting results for F 1s of different substrates after 50 cycles. (a) Cu. (b) $\text{Li}_2\text{S}@\text{Cu}$. (c) $\text{Li}_2\text{S}(111)@\text{Cu}$. The red percentages in the Figures represent the peak area ratio of Li-F/C-F.

Fig. R24 | The normalized XPS fitting results for S 2p of different substrates after 50 cycles. (a) Cu. (b) $\text{Li}_2\text{S}@Cu$. (c) $\text{Li}_2\text{S}(111)@Cu$.

Fig. R25 | Atomic percentages from the XPS survey spectra. (a) Cu. (b) $\text{Li}_2\text{S}@Cu$. (c) $\text{Li}_2\text{S}(111)@Cu$.

Table R1. Atomic percentages from the XPS survey spectra of Cu, Li₂S@Cu, Li₂S(111)@Cu substrates after 50 cycles.

		Atomic percentage								
Sputtering depth Element	Cu			Li ₂ S@Cu			Li ₂ S(111)@Cu			
	0 nm	10 nm	20 nm	0 nm	10 nm	20 nm	0 nm	10 nm	20 nm	
C	27.2	5.11	4.18	30.65	8.24	6.25	21.41	6.75	5.79	
O	23.37	41.1	40.68	28.08	41.07	40.62	27.45	38.92	37.79	
F	19.3	12.02	12.53	8.08	8.08	8.62	16.75	12.7	14.03	
Li	23.5	37.09	37.6	29.77	37.7	39.09	26.13	35.04	35.44	
S	6.29	4.03	4.04	3.2	4.21	4.56	3.97	4.29	4.38	

Comment 5: The influence of Li₂S crystal orientation on the Li transport dynamics was investigated by DFT calculations. More experimental evidences are suggested to supplement, e.g., the Li transference number and Li ion conductivity.

Response to Comment 5: We are grateful to the reviewer's insightful comments. According to the reviewer's valuable suggestion, we have conducted the related experiments. The transference number of Li⁺ (t^+) was evaluated using the Bruce-Vincent method¹⁷. The results were shown in **Fig. R26**, the symmetric cells with Cu, Li₂S@Cu and Li₂S(111)@Cu substrates exhibited the t^+ of 0.4, 0.64 and 0.68, respectively. The ionic conductivities with different substrates were also investigated. As shown in the Nyquist plots based on different substrates in **Fig. R27**, the calculated ionic conductivity of Cu, Li₂S@Cu and Li₂S(111)@Cu was 1.5×10^{-2} , 1.6×10^{-2} and 2.2×10^{-2} S cm⁻¹ at 25°C, respectively. The highest Li ion transfer number and Li ion conductivity with the Li₂S(111)@Cu substrate could be desirable to reduce the diffusion barrier and enhance the Li transport dynamics, which aligned well with our previous DFT

results (**Fig. 1**). In addition, the improved Li transport dynamics was highly beneficial for the battery high-rate performance (refer to the supplementary 3 C and 5 C tests with full cells in **Supplementary Figs. 63 and 67**).

Fig. R26| Steady-state current under 10 mV polarization for different substrates, inset shows EIS measurements before and after polarization. (a) Cu. (b) $\text{Li}_2\text{S}@Cu$. (c) $\text{Li}_2\text{S}(111)@Cu$.

Fig. R27| Nyquist plots of different substrates sandwiched by two stainless plates of steel.

Revision to Comment 5: The corresponding contents have been added in Supplementary Figs. 60-61.

The discussions have been added in the revised manuscript (page 18, line 19-22; page 19, line 1). The specific measurement procedures and calculated equations have been added in the *Methods* (page 26, line 7-13).

Comment 6: In Line 14, the authors claimed that the performance of full cell outperform the previously reported works, which is not fully accurate. The comparison should be better made by considering the N/P ratio, areal capacity, and current density. This requires careful revision in the revised manuscript.

Response to Comment 6: Thank you for your valuable comment. We have provided a detailed comparison according to your suggestion, which was showed in **Table R2**. Compared to previously reported work, the LFP and NCM811 full cells performance based on Li₂S(111)@Cu substrate exhibited better cycle stability under more stringent conditions.

Table R2. Comparison of electrochemical performance of full cells employing different substrates reported in literatures and this work.

Materials	Electrolytes	N/P ratio	Cathode areal capacity	Current density (C)	Lifespan	Ref
3D Cu-Zn substrate	1 M LiTFSI in DOL/DME with 1% LiNO ₃	3.3	0.3 mAh cm ⁻² LFP	0.5	300 cycles	18
Nanowires Cu ₂ S–Cu substrate	1 M LiTFSI in DOL/DME with 1% LiNO ₃	2	1 mAh cm ⁻² LFP	0.5	320 cycles	19
3D Cu@PDMS substrate	1 M LiTFSI in DOL/DME with 1% LiNO ₃	3	1 mAh cm ⁻² LFP	1	100	20
Ni-TAA/Cu substrate	1 M LiPF ₆ in EC/DEC (1/1)	5.9	0.85 mAh cm ⁻² LFP	1	400 cycles	21
CuO@Cu substrate	1 M LiTFSI in DOL/DME with 1% LiNO ₃	6	0.5 mAh cm ⁻² LFP	1	300 cycles	22
Sheet-like Cu@Li ₂ S substrate	1 M LiTFSI in DOL/DME with 1% LiNO ₃	3.3	1.8 mAh cm ⁻² LFP	1	200 cycles	23
Nanowire CuS substrate	1 M LiTFSI in DOL/DME with 2% LiNO ₃	7	N/A mAh cm ⁻² LFP	1	200 cycles	24

Co-2@NF substrate	1 M LiTFSI in DOL/DME with 1% LiNO ₃	14.1	0.42 mAh cm ⁻² LFP	2	480 cycles	25
CC-Ag substrate	1 M LiTFSI in DOL/DME with 1% LiNO ₃	15	0.33 mAh cm ⁻² LFP	4	30 cycles	26
cellulose/graphene carbon composite aerogel	1 M LiPF ₆ in 1:1 EC/DEC with 5% VC	Excess Li	1.42 mAh cm ⁻² LFP	5	1000	27
Fe-N@SSM substrate	1 M LiTFSI in DOL/DME with 2% LiNO ₃	3.33	1.8 mAh cm ⁻² LFP	5	280 cycles	28
GP substrate	1 M LiPF ₆ in 5:1 EMC/FEC	2	4 mAh cm ⁻² NCM532	0.3	160 cycles	29
Nanosheet Cu ₂ S@Cu substrate	1 M LiPF ₆ in EC/DEC/DMC (1:1:1)	2.5	4 mAh cm ⁻² NCM811	0.5	180 cycles	30
SNGO substrate	1 M LiPF ₆ in EC/DEC (1/1) with 1% VC+10% FEC	Excess Li infusion in host	1.7 mAh cm ⁻² NCM811	0.5	100 cycles	31
Li ₂ S coating	1 M LiPF ₆ in 1:1 EC/DEC	3.6	2.8 mAh cm ⁻² NCM532	1	100 cycles	32
Fe/LiF substrate	Advanced electrolyte: 2 M LiFSI in DME/BTFE (1/4)	1	3 mAh cm ⁻² NCM811	1	130 cycles	33
Nanorods Li₂S(111)@Cu substrate	1 M LiTFSI in DOL/DME with 2% LiNO₃	4	1 mAh cm⁻², LFP	0.5	400 cycles	This work
		10	1 mAh cm⁻², LFP	3	1150 cycles	
		10	1 mAh cm⁻², LFP	5	2060 cycles	
	1 M LiPF₆ in EC/DEC with 5% FEC	1.3	4 mAh cm⁻² NCM811	0.3	160 cycles	
		2.5	4 mAh cm⁻² NCM811	3	163 cycles	
		2.5	4 mAh cm⁻² NCM811	5	177 cycles	

Revision to Comment 6: The Table R2 has been updated in the Supplementary Table 6. The

corresponding discussions have been added in page 20, line 22-23; page 21, line 1.

Comment 7: In Figure 4b, it is difficult to see the detailed Coulombic efficiencies (CEs). All the plots containing the CE results should be rescaled or supplemented with an inset so that any fluctuations around 100% CE can be clearly observed. Further, for the determination of the electrochemical data (capacity, CE, cycling stability), multiple determinations should be performed and error bars should be provided to fully verify the differences between $\text{Li}_2\text{S}(111)\text{@Cu}$ and bare Cu.

Response to Comment 7: Thank you very much for your careful check and valuable suggestions. We have rescaled all the CE results and provided parallel experiments to fully demonstrate the different between the $\text{Li}_2\text{S}(111)\text{@Cu}$ and Cu substrates. The specific details are as follows.

Rescaled CE data: The CE data in **Fig. 4b**, **Fig. 5c-5e**, **Supplementary Figs. 47-48**, **62a** and **65a** have been rescaled and displayed in **Figs. R28-R34** to better visualize any fluctuations.

Fig. R28| Rescaled CE data of Fig. 4b. CE versus cycle number under a plating-full-stripping mode at 0.5 mA cm^{-2} with 0.5 mAh cm^{-2} , the insets are SEM images of the substrates after 50 cycles.

Fig. R29| Rescaled CE data of Supplementary Fig. 47. CE test of the $\text{Li}_2\text{S}@\text{Cu}$ substrate cell was conducted in a plating-full-stripping mode, and the inset features a SEM image of the substrate structure after 50 cycles.

Fig. R30| Rescaled CE data of Supplementary Fig. 48. CE versus cycle number of Cu and $\text{Li}_2\text{S}(111)@\text{Cu}$ substrate cells at 1 mA cm^{-2} and 1 mAh cm^{-2} .

Fig. R31| Rescaled CE data of Fig. 5c. Discharge capacity retention of Li-substrate||LFP full cells at 0.5 C rate.

Fig. R32| Rescaled CE data of Supplementary Fig. 62a. The electrochemical performance of LFP full cell based on $\text{Li}_2\text{S}@Cu$ substrate. Discharge capacity retention of Li-substrate||LFP full cells at 0.5 C rate.

Fig. R33| Rescaled CE data of Fig. 5e. Discharge capacity retention of Li-substrate||NCM811 full cells at 0.3 C rate.

Fig. R34| Rescaled CE data of Supplementary Fig. 65a. The electrochemical performance of the NCM811 full cell based on $\text{Li}_2\text{S}@Cu$ substrate. Discharge capacity retention of Li-substrate||NCM811 full cells at 0.3 C rate.

Parallel experiments: Multiple measurements of the decisive electrochemical performance of half cells

and full cells based on the Cu and Li₂S(111)@Cu substrates have been conducted, and the results were shown in **Figs. R35-R37**. CE has been widely used as an important quantifiable indicator for the efficiency and reversibility of batteries. **Fig. R35** exhibited the parallel experiments of CE test under a partial-stripping mode at 1 mA cm⁻² with 1 mAh cm⁻², the mean CE based on the Cu and Li₂S(111)@Cu substrates were 97.5% and 99.2%, respectively. **Fig. R36** showed the parallel experiments of CE versus cycle number under a plating-full-stripping mode at 0.5 mA cm⁻² with 0.5 mAh cm⁻². The variation of performances in the parallel experiments displayed the same tendency with the results in the manuscript, demonstrating the good reproducibility of half cells. To further verify the performance of two substrates in full cells, parallel experiments were carried out by evaluating full cells based on Cu and Li₂S(111)@Cu as anode substrates and commercial NMC811 as cathodes. Since the NMC811 cathodes were commercially available, the differences of the substrates could be highlighted, the results as shown in **Fig. R37**. In parallel experiments, the NCM 811 full cell based on the Li₂S(111)@Cu substrate still demonstrated steady capacities, with 80% capacity retentions over 148 cycles (parallel 1) and 184 cycles (parallel 2). Meanwhile, the average CEs were 99.6% and 99.5% in parallel 1 and 2, respectively. However, for the Cu substrate, the rapid degradations were observed in repeated experiments, with 80% capacity retentions only after 46 (parallel 1) and 20 cycles (parallel 2). In summary, the mean cycle numbers with 80% capacity retention for Cu and Li₂S(111)@Cu substrates were 38 and 164 cycles, with the standard deviations of 16 and 18, respectively. The corresponding mean average CE were 98.6% and 99.6%, with the standard deviations of 0.29 and 0.1, respectively. Therefore, the repeated experiments fully demonstrated the improvement of CE, capacity and lifespan by Li₂S(111)@Cu substrate.

Fig. R35 | CE of Li||substrate half-cells with Cu and Li₂S(111)@Cu substrates. (a) Voltage profiles of CE test under a partial-stripping mode at 1 mA cm⁻² with 1 mAh cm⁻². (b) CE values. Error bars represent s.d.

Fig. R36 | Parallel experiments of CE test under a plating-full-stripping mode at 0.5 mA cm⁻² with 0.5 mAh cm⁻². (a) CEs versus cycle number. (b) CE versus cycle number. Error bars represent s.d. (c) CE versus cycle number of parallel experiment 1. (d) CE versus cycle number of parallel experiment 2.

Fig. R37| Parallel experiments of NCM811 full cells based on Cu and Li₂S(111)@Cu substrates. (a) Discharge capacity retention at 0.3 C rate. (b) Cycle number for 80% capacity retention and average CE values. Error bars represent s.d. (c) Discharge capacity retention at 0.3 C rate of parallel experiment 1. (d) Corresponding GDC profiles of parallel full cell 1 at different cycles. (e) Discharge capacity retention at 0.3 C rate of parallel experiment 2. (f) Corresponding GDC profiles of parallel full cell 2 at different cycles.

Revision to Comment 7: The corresponding rescaled Figures have been updated in Fig. 4b, Fig. 5c-5e,

Supplementary Figs. 47-48, 62a and 65a. The contents of parallel experiments have been added in

Supplementary Figs. 44, 46 and 65. The contents of parallel experiments have been added in the revised

manuscript (page 15, line 13; page 16, line 2-3; page 20, line 7-8).

Comment 8: In Figures 2a and 3k, the schematic illustration of the highly orientated Li₂S arrays should

re-draw according to the true morphology of the Li_2S arrays in Figure 3f. The current version will cause serious confusion and misunderstanding.

Response to Comment 8: Thank you for your suggestion, we feel sorry for any confusion in our previous schematic. We have re-drawn the corresponding nanorod arrays in Fig. 2a, Fig.3k and Supplementary Fig. 43, as shown in Fig. R38, Fig. R39 and Fig. R40, respectively.

Fig. R38| Schematic representation for the design of $\text{Li}_2\text{S}(111)\text{@Cu}$ substrate.

Fig. R39| Schematic illustration of Li deposited on Cu and $\text{Li}_2\text{S}(111)\text{@Cu}$ substrates.

Fig. R40| A schematic illustration depicting the loose growth of Li deposited on $\text{Li}_2\text{S@Cu}$ substrate.

Revision to Comment 8: The corresponding re-drew schematic illustrations have been updated in Fig. 2a

and Fig. 3k of the revised manuscript, and Supplementary Fig. 43 of the revised Supplementary Information.

Reviewer #3: *In this study, the authors engineered three-dimensional vertically aligned $\text{Li}_2\text{S}(111)\text{@Cu}$ nanorod arrays as substrates for Li anodes. They manipulated the crystal orientation of Li_2S using sulfur-defect engineering to enhance Li^+ transport and suppress dendritic growth. However, the study lacks depth in its exploration of material fabrication mechanisms and battery performance evaluations. I recommend substantial revisions before considering acceptance. Additionally, the authors should address the following points:*

Response: We would like to express our sincere appreciation for your positive comments and the valuable feedback to help us deeply delve the material synthesis mechanisms and the evaluation of battery performance. We have carefully considered each question and attached a point-by-point response as follows.

Comment 1: *The authors prepared S-deficient $\text{Cu}_2\text{S}@Cu$ NRs by annealing in an argon atmosphere, resulting in a small-angle shift in XRD characteristic peaks. They need to further explain the annealing-induced S-deficiency and the associated peak shifts.*

Response to Comment 1: Thank you for your comment. We would like to response in the following two aspects:

Explanation for the annealing-induced S-deficiency: Annealing has been developed as a vital strategy for defect engineering, e.g., S defects are easily induced during the Ar annealing process^{34,35}.

Explanation for the peak shifts in XRD patterns: After introducing S defects, the characteristic peaks of Cu_2S shifted to lower angle because the expansion in the lattice³⁶. Specifically, the removal of S gave rise

to coordinately unsaturated metal centers, resulting in electrostatic repulsion between positively charged S defects and the surrounding Cu^+ cations, and the subsequent lattice expansion³⁷. Therefore, the XRD peaks of Cu_2S_x sample shifted to lower 2θ values according to the Bragg's Law³⁸.

Revision to Comment 1: The corresponding contents of the annealing-induced S-deficiency have been added in the revised manuscript in the *Methods* (page22, line23; page 23, line 1). The explanation for the peak shifts in XRD patterns has been added below the corresponding XRD patterns (Supplementary Fig. 4).

Comment 2: In Figure 5c, for full battery testing, the battery using Cu as the anode substrate and LFP as the cathode showed slightly lower discharge capacity but better cycling stability compared to the $\text{Li}_2\text{S}(111)\text{@Cu}$ -based battery. However, the abrupt termination after 200 cycles needs clarification.

Response to Comment 2: Thank you for raising this issue. We would like to clarify that the test program was terminated at 200 cycles because we just felt the much lower discharge capacity of LFP full cell based on the Cu substrate could sufficiently exhibit the difference between the Cu and $\text{Li}_2\text{S}(111)\text{@Cu}$ substrates. We are sorry for the careless. We have conducted more experiments to evaluate the cycling performance of LFP full cell based on Cu substrate. As shown in **Fig. R41**, the three LFP full cells based on Cu1, Cu2 and Cu3 substrates displayed similar capacity retentions of 73%, 74% and 74% after 200 cycles at 0.5 C, respectively. The average CEs based on these Cu substrates were around 99.17%, 99.18% and 99.26%, respectively. We apologize for the confusion, we have updated Cu1 to the revised manuscript to clarify the performance lies between the Cu and the $\text{Li}_2\text{S}(111)\text{@Cu}$ based LFP full cells (**Fig. R42**).

Fig. R41| Discharge capacity retention of LFP full cells base on Cu substrates at 0.5 C rate.

Fig. R42| Electrochemical performance of LFP full cells based on Cu and $\text{Li}_2\text{S}(111)\text{@Cu}$ substrates. (a) Discharge capacity retention of Li-substrate||LFP full cells at 0.5 C rate. (b) Corresponding GDC profiles of LFP full cells at different cycles.

Revision to Comment 2: The corresponding discussion have been added in the revised manuscript (page 19, line 10-11) and the corresponding contents have been added in Fig. 5c-5d.

Comment 3: While the authors demonstrated excellent cycle stability and high Coulombic efficiency in full cells, practical applications typically employ pouch cells. It is suggested that they extend their evaluations to high-load pouch cells to more convincingly demonstrate the performance and practical potential of the material.

Response to Comment 3: We would like to express our sincere appreciation for the valuable suggestion.

We agreed with the reviewer's comments that pouch cells are more convincing in demonstrating the

practical application of our research. As a results, pouch cell (100 mAh, $7.5 \times 7.5 \text{ cm}^2$) with high-loading LFP (20 mg cm^{-2}) as cathode, $\text{Li}_2\text{S}(111)\text{@Cu}$ substrate with Li (3 mAh cm^{-2}) as anode, electrolyte (3.3 g Ah^{-1}) and N/P ratio of 1 were assembled. **Fig. R43a** shown the schematic illustration of the LFP pouch cell. The performance of LFP pouch cell was evaluated at 0.33 C charge and discharge, and the voltage range was 2.5-3.85 V. **Fig. R43b** exhibited the voltage-capacity profiles at different cycles, the larger polarization voltage compared to that of coin cell was due to the high-loading LFP used in the pouch cell. **Fig. R43c** showed the cycling performance of the LFP pouch cell, which demonstrated a maximum discharge capacity of 99.96 mAh and maintained a capacity retention of 83% after 200 cycles, with the average CE of 99.88%. The superior cycling stability and high CE in pouch cell, further demonstrated the practical application potential of $\text{Li}_2\text{S}(111)\text{@Cu}$ substrate.

Fig. R43| Electrochemical performance of LFP pouch cell based on the $\text{Li}_2\text{S}(111)\text{@Cu}$ substrate. (a) Schematic illustration of the LFP pouch cell. (b) Corresponding GDC profiles of the LFP pouch cell at different cycles. (c) Cycling performance of the LFP pouch cell at 0.33 C.

Revision to Comment 3: The corresponding contents have been added in Supplementary Fig. 68. The discussions have been added in the revised manuscript (page 21, line 1-4; page 25, line 7-8).

Comment 4: To further validate the kinetic enhancements in lithium deposition and exfoliation provided by the $\text{Li}_2\text{S}(111)\text{@Cu}$ anode substrate, it is advisable to conduct tests using symmetric or half-cell configurations at high current densities (3 or 5 mAh cm^{-2}).

Response to Comment 4: We appreciate the reviewer's suggestions about validating the kinetic enhancements in Li plating/stripping at high current densities. Following your valuable suggestions, we performed the half-cells testing with Cu and $\text{Li}_2\text{S}(111)\text{@Cu}$ substrates under challenging conditions of high current density (3 or 5 mA cm^{-2}) and high capacity (3 and 5 mAh cm^{-2}). As shown in **Fig. R44**, the CE tests were carried out under a plating-full-stripping mode, the half-cell based on $\text{Li}_2\text{S}(111)\text{@Cu}$ substrate delivered high average CEs (>99.19%) for 165 cycles at 3 mA cm^{-2} with 3 mAh cm^{-2} , while the CE of the cell based on Cu substrate rapidly deteriorated only after 25 cycles. Even under a higher capacity of 5 mAh cm^{-2} and a higher current density of 5 mA cm^{-2} , the half-cell based on $\text{Li}_2\text{S}(111)\text{@Cu}$ could still maintain stable CEs over 55 cycles, in stark contrast to the drastic fluctuations in CE observed with Cu substrate (**Fig. R45**). In summary, the CE of half-cells based on $\text{Li}_2\text{S}(111)\text{@Cu}$ substrate outperformed that with Cu substrate even at high current capacities and high current densities, further demonstrating the effective Li plating and stripping.

Fig. R44 | CE of Li||substrate half-cells with Cu and $\text{Li}_2\text{S}(111)\text{@Cu}$ substrates under a plating-full-stripping mode at high current density of 3 mA cm^{-2} with 3 mAh cm^{-2} .

Fig. R45 | CE of Li||substrate half-cells with Cu and $\text{Li}_2\text{S}(111)\text{@Cu}$ substrates under a plating-full-stripping mode at high current density of 5 mA cm^{-2} with 5 mAh cm^{-2} .

Revision to Comment 4: The corresponding contents have been added in Supplementary Figs. 49-50.

The discussions have been added in the revised manuscript (page 16, line 17-21).

Comment 5: Considering that power batteries in practical applications often operate at 3 C or 5 C rates, the current tests with LiFePO_4 and NCM811 cathodes at only 0.5 C and 0.3 C are insufficient. The authors should consider conducting high-rate discharge tests to meet contemporary demands.

Response to Comment 5: Thank you for your important suggestion. We have conducted the full cell testing with LFP and NCM811 cathodes at the higher rates of 3 C and 5 C. As shown in **Fig. R46a**, the LFP full cell based on the $\text{Li}_2\text{S}(111)\text{@Cu}$ substrate displayed a steady capacity at a high rate of 3 C, with

capacity retentions of 80% after 1150 cycles, and 75% after 1500 cycles. However, the full cell based on Cu substrate exhibited a pronounced deterioration only after 54 cycles. **Fig. R46b** showed the galvanostatic discharge/charge (GDC) curves of LFP full cells during cycling at 3 C. The polarization voltage of the cell based on the $\text{Li}_2\text{S}(111)@\text{Cu}$ substrate was 199 mV, while that of Cu substrate was as large as 361 mV, indicating the slow Li transport dynamics. Even at a high rate of 5 C, the LFP full cell based on the $\text{Li}_2\text{S}(111)@\text{Cu}$ substrate still maintained high stability for 2060 cycles with a capacity retentions of 80%, and maintained 73% capacity after 2500 cycles (**Fig. R46c**). Incidentally, the capacity fluctuations observed in later cycles may result from the variations in internal properties during long-term cycling as well as fluctuations in the ambient lab temperature³⁹. Meanwhile, the LFP full cell based on Cu substrate still experienced rapid degradation, and the average CE was as low as 69.0% during the first 50 cycles, followed by significant fluctuations. The GDC curves of Cu substrate in **Fig. R46d** revealed obvious overcharging behavior and large polarization voltage of 393 mV. In stark contrast to Cu substrate, the full cell based on the $\text{Li}_2\text{S}(111)@\text{Cu}$ substrate demonstrated controlled polarization voltage of 252 mV. Furthermore, the high-rate tests of the Cu and $\text{Li}_2\text{S}(111)@\text{Cu}$ substrates were tested in the full cell configuration with commercial high-loading NCM811 cathodes, the results as shown in **Fig. R47**. The NCM811 full cell based on the $\text{Li}_2\text{S}(111)@\text{Cu}$ substrate delivered an initial capacity of 159.5 mAh g⁻¹ at 3 C, and presented a capacity retention of 80% after 163 cycles with the average CE of 99.4%. While that of Cu substrate displayed an initial capacity of 107.2 mAh g⁻¹, and a rapid capacity decay after 17 cycles with a low average CE of 92.79% (**Fig. R47a**). The corresponding charge/discharge profiles of various cycles were presented in **Fig. R47b**, which also illustrated the rapid degradation of the NCM811 full cell based on Cu substrate and the superior stability of $\text{Li}_2\text{S}(111)@\text{Cu}$ substrate. We further demonstrated the

high-rate performance of the $\text{Li}_2\text{S}(111)\text{@Cu}$ substrate at 5 C, as shown in **Fig. R47c**, the maximum discharge capacity of the NCM811 full cell based on the $\text{Li}_2\text{S}(111)\text{@Cu}$ substrate measured 90.9 mAh g^{-1} , and maintained 80% capacity retention after 177 cycles with the average CE of 99.1%. For the Cu substrate, the NCM811 full cell deteriorated quickly, with capacity dropping from 76.1 to 61 mAh g^{-1} (80% retention) within only 7 cycles, and the average CE as low as 69.0%. The corresponding GDC curves in **Fig. R47d** showed significant voltage polarization in Cu substrate. In summary, the observed stability cycle and low voltage polarization for the $\text{Li}_2\text{S}(111)\text{@Cu}$ substrate in high-rate tests with both the LFP and NCM811 full cells, indicating the effective Li plating/stripping and fast Li transfer kinetics. These results collectively demonstrated the potential of the $\text{Li}_2\text{S}(111)\text{@Cu}$ substrate in high-rate applications.

Fig. R46| Electrochemical performance of LFP full cells based on Cu and $\text{Li}_2\text{S}(111)\text{@Cu}$ substrates at high rates. (a) Discharge capacity retention of Li-substrate||LFP full cells at 3 C charge and discharge rate. (b) Corresponding GDC profiles of LFP full cells in Fig. R44a at various cycles. (c) Discharge capacity retention of Li-substrate||LFP full cells at 5 C charge and discharge rate. (d) Corresponding GDC profiles of LFP full cells in Fig. R44c at various cycles.

Fig. R47| Electrochemical performance of NCM811 full cells based on Cu and Li₂S(111)@Cu substrates at high rates. (a) Discharge capacity retention of Li-substrate||NCM811 full cells at 3 C discharge rate and 0.33 C charge rate. (b) Corresponding GDC profiles of NCM811 full cells at various cycles under 3 C discharge rate and 0.33 C charge rate. (c) Discharge capacity retention of Li-substrate||LFP full cells at 5 C discharge rate and 0.5 C charge rate. (d) Corresponding GDC profiles of NCM811 full cells at various cycles under 5 C discharge rate and 0.5 C charge rate.

Revision to Comment 5: The corresponding contents have been added in Supplementary Figs. 63 and 67.

The discussions have been added in the revised manuscript (page 19, line 12-17; page 20, line 12-20).

Reference

- 1 McDowell, M. T. et al. In situ observation of divergent phase transformations in individual sulfide nanocrystals. *Nano Lett.* **15**, 1264-1271 (2015).
- 2 Zhang, X. et al. Hidden vacancy benefit in monolayer 2D semiconductors. *Adv. Mater.* **33**, 2007051 (2021).
- 3 Wang, X. et al. Single-atom vacancy defect to trigger high-efficiency hydrogen evolution of MoS₂. *J. Am. Chem. Soc.* **142**, 4298-4308 (2020).
- 4 Lei, F. et al. Oxygen vacancies confined in ultrathin indium oxide porous sheets for promoted visible-light water splitting. *J. Am. Chem. Soc.* **136**, 6826-6829 (2014).
- 5 Buerger, M. J., Bernhardt, J. W. Distribution of atoms in high chalcocite, Cu₂S. *Science* **141**, 276-277 (1963).
- 6 Li, Y., Romero, N. A. and Lau, K. C. Structure-property of lithium-sulfur nanoparticles via molecular dynamics simulation. *ACS Appl. Mater. Interfaces.* **10**, 37575-37585 (2018).
- 7 Kalimuldina, G. et al. Morphology and dimension variations of copper sulfide for high-performance electrode in rechargeable batteries: A Review. *ACS Appl. Energy Mater.* **3**, 11480-11499 (2020).
- 8 Boebinger, M. G. et al. Distinct nanoscale reaction pathways in a sulfide material for sodium and lithium batteries. *J. Mater. Chem. A.* **5**, 11701-11709 (2017).
- 9 Xie, C. et al. Insight into the design of defect electrocatalysts: From electronic structure to adsorption energy. *Mater. Today.* **31**, 47-68 (2019).
- 10 Zhang, S. et al. Configuration regulation of active sites by accurate doping inducing self-adapting defect for enhanced photocatalytic applications: A review. *Chem. Rev.* **478**, 214970 (2023).
- 11 Heiskanen, S. K., Kim, J. and Lucht, B. L. Generation and evolution of the solid electrolyte interphase of lithium-ion batteries. *Joule.* **3**, 2322-2333 (2019).
- 12 Wan, J. et al. Single-ion conducting interlayers for improved lithium metal plating. *Energy Stor. Mater.* **63**, 103029 (2023).
- 13 Chen, C., Liang, Q., Wang, G., Liu, D. and Xiong, X. Grain-boundary-rich artificial SEI layer for

- high-rate lithium metal anodes. *Adv. Funct. Mater.* **32**, 2107249 (2021).
- 14 Li, G. *et al.* Organosulfide-plasticized solid-electrolyte interphase layer enables stable lithium metal anodes for long-cycle lithium-sulfur batteries. *Nat. Commun.* **8**, 850 (2017).
- 15 Yang, J. *et al.* Prolongating cycling lifetime of lithium metal batteries with monolithic and inorganic-rich solid electrolyte interphase. *Energ. Environ. Sci.* **16**, 3837-3846 (2023).
- 16 Zhou, G., Paek, E., Hwang, G. S. and Manthiram, A. Long-life Li/polysulphide batteries with high sulphur loading enabled by lightweight three-dimensional nitrogen/sulphur-codoped graphene sponge. *Nat. Commun.* **6**, 7760 (2015).
- 17 Evans, J., Vincent, C. A., and Bruce, P. G. Electrochemical measurement of transference numbers in polymer electrolytes. *Polymer*. **28**, 2324-2328 (1987)..
- 18 Yun, Q. *et al.* Chemical dealloying derived 3D porous current collector for Li metal anodes. *Adv. Mater.* **28**, 6932-6939 (2016).
- 19 Zhai, P. *et al.* In situ generation of artificial solid-electrolyte interphases on 3D conducting scaffolds for high-performance lithium-metal anodes. *Adv. Energy Mater.* **10**, 1903339 (2020).
- 20 Wang, X. *et al.* Stress-driven lithium dendrite growth mechanism and dendrite mitigation by electroplating on soft substrates. *Nat. Energy*. **3**, 227-235 (2018).
- 21 Ke, S.-W. *et al.* Redox-active covalent organic frameworks with nickel-bis(dithiolene) units as guiding layers for high-performance lithium metal batteries. *J. Am. Chem. Soc.* **144**, 8267-8277 (2022).
- 22 Liu, Y. *et al.* Integrated gradient Cu current collector enables bottom-up Li growth for Li metal anodes: role of interfacial structure. *Adv. Sci (Weinh)*. **10**, 2301288-2301295 (2023).
- 23 Zhang, R. *et al.* Decreasing interfacial pitfalls with self-grown sheet-like Li₂S artificial solid-electrolyte interphase for enhanced cycling performance of lithium metal anode. *Small*. **19**, 2208095 (2023).
- 24 Zou, P., Wang, C., Qin, J., Zhang, R. & Xin, H. L. A reactive wetting strategy improves lithium metal reversibility. *Energy Stor. Mater.* **58**, 176-183 (2023).
- 25 Guo, C. *et al.* Uniform lithiophilic layers in 3D current collectors enable ultrastable solid

- electrolyte interphase for high-performance lithium metal batteries. *Nano Energy*. **96**, 107121 (2022).
- 26 Tian, R. et al. Oriented growth of Li metal for stable Li/carbon composite negative electrode. *Electrochim. Acta*. **292**, 227-233 (2018).
- 27 Mao, H. et al. Current-density regulating lithium metal directional deposition for long cycle-life Li metal batteries. *Angew. Chem. Int. Ed.* **60**, 19306-19313 (2021).
- 28 u, X., Duan, H., Zhang, L., Hu, Y. and Deng, Y. A 3D framework with an in situ generated Li₃N solid electrolyte interphase for superior lithium metal batteries. *Adv. Funct. Mater.* **33**, 2308022 (2023).
- 29 Li, Y. et al. Artificial graphite paper as a corrosion-resistant current collector for long-life lithium metal batteries. *Adv. Funct. Mater.* **33**, 2214523 (2023).
- 30 Yang, Z. et al. Ultra-smooth and dense lithium deposition toward high-performance lithium metal batteries. *Adv. Mater* **35**, 2210130 (2023).
- 31 Ni, S. et al. A 3D framework with Li₃N-Li₂S solid electrolyte interphase and fast ion transfer channels for a stabilized lithium-metal anode. *Adv. Mater* **35**, 2209028 (2023).
- 32 Chen, H. et al. Uniform high ionic conducting lithium sulfide protection layer for stable lithium metal anode. *Adv. Energy Mater* **9**, 1900858 (2019).
- 33 Wu, Z. et al. Growing single-crystalline seeds on lithiophobic substrates to enable fast-charging lithium-metal batteries. *Nat. Energy* **8**, 340-350 (2023).
- 34 Xu, J. et al. Frenkel-defected monolayer MoS₂ catalysts for efficient hydrogen evolution. *Nat. Commun.* **13**, 2193 (2022).
- 35 You, M. et al. High temperature induced S vacancies in natural molybdenite for robust electrocatalytic nitrogen reduction. *J. Colloid Interface Sci.* **599**, 849-856 (2021).
- 36 Liu, X. et al. Uncovering the effect of lattice strain and oxygen deficiency on electrocatalytic activity of perovskite cobaltite thin films. *Adv Sci (Weinh)* **6**, 1801898, (2019).
- 37 Qian, K. et al. Elucidating the strain-vacancy-activity relationship on structurally deformed Co@CoO nanosheets for aqueous phase reforming of formaldehyde. *Small* **17**, e2102970, (2021).

- 38 Liu, X. et al. The lattice expansion in nanometre-sized Ni polycrystals. *J. Phys.: Condens. Matter.* **6**, 497-502 (1994).
- 39 Krieger, E. M., Cannarella, J. and Arnold, C. B. A comparison of lead-acid and lithium-based battery behavior and capacity fade in off-grid renewable charging applications. *Energy.* **60**, 492-500 (2013).

Sulfur defect engineering controls Li_2S crystal orientation for dendrite-free lithium metal batteries

Authors: Jin-Xia Lin[#], Peng Dai[#], Sheng-Nan Hu[#], Shi-Yuan Zhou, Gyeong-Su Park, Chen-Guang Shi, Jun-Fei Shen, Yu-Xiang Xie, Wei-Chen Zheng, Hui Chen, Shi-Shi Liu, Hua-Yu Huang, Ying Zhong, Jun-Tao Li, Xiaoyang (Jerry) Huang, Rena Oh^{*}, Wen-Feng Lin^{*}, Ling Huang^{*}, Shi-Gang Sun^{*}

We would like to express our sincere gratitude to the editor and all the three reviewers for the valuable feedback and points on our work. The comments raised by the reviewer have helped us improve the quality of the manuscript and spark thoughts about future work. In the detailed point-by-point response to each comment below, the reviewer comments are laid out in *blue italicized font* and our responses are given in normal font for clarify. Additionally, any changes in the manuscript and supplementary information are highlighted **in yellow**.

Thank you once again to the editor and all the three reviewers for their important comments.

Responses to Reviewers:

To Reviewer #1	2
To Reviewer #2 (Fig. R1):	3
To Reviewer #3 (Figs. R2-R3):	4-8
Reference	9

A Detailed Point-by-Point Responses to Reviewers' Comments

Reviewer #1: *The quality of the revised article has been improved, and it is recommended to accept it.*

Response: Thank you very much for your positive feedback and approval of our revised article.

Reviewer #2: *In the revised manuscript, the authors have addressed my comments and suggestions, and the work has been significantly improved. I would like to recommend its publication in Nature Communications.*

A minor error should be revised: in Supplementary Fig. 65a, the unit of y axis should be mAh/g, rather than mAh/cm².

Response: We are very grateful to the reviewer's positive comments. We apologize for the careless error, the unit of y axis in Supplementary Fig. 65a has been corrected. Thank you very much for your careful reading.

Fig. R1| Re-draw Supplementary Fig. 65a. The electrochemical performance of the NCM811 full cell based on Li₂S@Cu substrate. Discharge capacity retention of Li-substrate||NCM811 full cells at 0.3 C rate.

Reviewer #3: *The authors' efforts to include additional experiments are commendable. The revised manuscript has shown significant improvement, and I recommend its publication in Nature Communications after addressing the following issues:*

Response: We sincerely appreciate your further valuable feedback. Your insightful comments definitely enhance the quality of this work. To address your concerns, we have carefully considered each comment and attached a point-by-point response as follows.

Comment 1: *Why was Li₂S chosen as the artificial SEI? Many studies have focused on inducing uniform lithium deposition through crystal surface engineering on Cu substrates. How does this work provide new insights or innovations compared to previous research?*

Response to Comment 1: We are thankful to the reviewer's comment. We believe there may have been a misunderstanding of the scope of our study. We feel sorry for any confusion in our previous description. In this study, we constructed Li₂S(111)@Cu nanorod arrays as a Li anode substrate, Li₂S was chosen as a component of the anode substrate rather than the artificial SEI. Cu as the comment substrate plays a vital role in the electron transfer, while the lithiophobicity nature of Cu could accelerate the Li dendritic growth. We investigated the Li₂S material due to its high Li affinity and high Li ion conductivity^{1,2}, which can facilitate the Li transport at Li/substrate interface. We appreciate the reviewer for this input, we have improved our manuscript to clarify that in the "Introduction" section.

We agree with the reviewer's comment that studies have focused on inducing uniform lithium deposition through crystal surface engineering on Cu substrates³⁻⁷. These reports reveal the relationship

between the anion adsorption ability³, Li adsorption and migration energy^{4,5}, surface lattice⁶ and deposition structure⁷ with Cu crystal orientation. Different from these studies, our work identified a substrate-dependent Li nucleation process and a facet-dependent growth mode, revealed the profound impact of electrode substrate design and engineering on Li nucleation and growth behavior, that is essential for achieving dendrite-free Li deposition. The comprehensive investigation into the impact of Li₂S at the Li/substrate interface and its exposed crystallographic facets on Li deposition, offering new insights into this area. We also want to emphasize that our work introduced a novel approach by employing S defect engineering to control Li₂S crystal orientation, and conducted an in-depth investigation into the mechanism of how S defects in Cu₂S_x@Cu can induce the crystal facet selectivity of Li₂S. We uniquely observed lithiated Cu₂S_x with S defects by using advanced techniques such as HR-STEM/iDPC and HR-STEM-EELS to effectively characterize light elements like S and Li atoms, detected previously unidentified phases. These insights could play a vital role in designing advanced Li anode substrates. Therefore, we consider our work demonstrated innovation in the elucidation of structure-performance relationships, the comprehensive analysis of theoretical models and the engineering design of materials.

We hope our response can address your concerns and meet your expectations. Thank you again for your valuable feedback.

Revision to Comment 1: The corresponding contents have been added in the revised manuscript (page 3, line 10-11).

Comment 2: I would appreciate a determination of the mass proportion of each component in the full cells,

with particular emphasis on the artificial SEI, as this information is essential for assessing the commercial potential of this work.

Response to Comment 2: Thank you for providing your important suggestion. We have supplemented the mass proportion of each component in the pouch cell. As we clarified in Comment 1, our work focused on the impact of Li anode substrate design and engineering on Li nucleation and growth behavior. Therefore, we consider that the mass proportion particularly emphasized in your comment pertains to the $\text{Li}_2\text{S}(111)\text{@Cu}$ substrate in full cell. As shown in **Table R1**, the $\text{Li}_2\text{S}(111)\text{@Cu}$ substrate with Li as anode accounts 31.58% of the total mass of the full cell. This high percentage mainly due to the heavy weight of the Cu foam used during the synthesis, and the mass of $\text{Li}_2\text{S}(111)\text{@Cu}$ is only 6.67% heavier than Cu foam. Thus, we believe that by replacing the Cu foam with lighter Cu foil, the $\text{Li}_2\text{S}(111)\text{@Cu}$ foil substrate could be commercially viable. We appreciate the reviewer for raising this important point, which will be further investigated in our follow up studies.

Table R1. The mass proportion of each component in the LFP pouch cell.

Cell component	Specification	Parameters	Mass proportion
LFP Cathode with Al substrate	Active material mass loading	20 mg cm ⁻²	12.42%
	Active material ratio	90.93%	
	Number	1	
	Weight	0.795 g	
Anode ($\text{Li}_2\text{S}(111)\text{@Cu}$ substrate with 3 mAh cm ⁻² Li)	Weight	2.022 g	31.58%
	Number	1	
Electrolyte	Weight	0.386 g	6.03%
Separator	Weight	0.045 g	0.70%
Package	Weight	2.798 g	43.70%
Tabs	Weight	0.357 g	5.57%
Total	Weight	6.403 g	100%

Revision to Comment 2: The corresponding contents and discussions have been added in the revised

manuscript (page 25, line 10-11) and Supplementary Table 7.

Comment 3: I noted that the authors have added a cycling test for pouch cells. Can the morphology of the artificial SEI be maintained after cycling? It is recommended to include SEM and other relevant characterizations to confirm this.

Response to Comment 3: Thank you for your suggestion. As we mentioned in **Response to Comment 1**, this study introduces the $\text{Li}_2\text{S}(111)\text{@Cu}$ material as Li anode substrate rather than artificial SEI. We feel sorry for the confusion. In the insets of **Fig. 4a** and **Fig. 4b**, SEM images displayed the morphology and structural integrity of $\text{Li}_2\text{S}(111)\text{@Cu}$ substrate after Li plating/stripping. In the full cell configuration, the $\text{Li}_2\text{S}(111)\text{@Cu}$ substrate with pre-deposited planar Li served as anode. After cycling, the SEM image showed the deposited Li on the $\text{Li}_2\text{S}(111)\text{@Cu}$ substrate maintained an even morphology (**Fig. R2a**). After Li stripping, the $\text{Li}_2\text{S}(111)\text{@Cu}$ substrate kept good nanorod array structure (**Fig. R2b**). TEM measurements were also carried out to observe the structure of the cycled substrate, further demonstrating the morphology integrity of the $\text{Li}_2\text{S}(111)\text{@Cu}$ substrate (**Fig. R3**). We hope these clarifications address the reviewer's concerns. Thank you again for your valuable feedback.

Fig. R2| SEM images after 100 cycles. (a) The cycled Li anode of the $\text{Li}_2\text{S}(111)\text{@Cu}$ with deposited Li.

(b) The cycled $\text{Li}_2\text{S}(111)\text{@Cu}$ substrate.

Fig. R3| TEM images of the cycled $\text{Li}_2\text{S}(111)\text{@Cu}$ substrate.

Revision to Comment 3: The corresponding contents and discussions have been added in the revised manuscript (page 21, line 4-6) and Supplementary Fig. 69.

Reference

1. Jand, S. P., Zhang, Q. & Kaghazchi, P. Theoretical study of superionic phase transition in Li_2S . *Sci. Rep.* **7**, 5873 (2017).
2. Chen, H. et al. Uniform high ionic conducting lithium sulfide protection layer for stable lithium metal anode. *Adv. Energy Mater.* **9**, 1900858 (2019).
3. Hao, Z. et al. The Dependence of Solid Electrolyte Interphase on the Crystal Facet of Current Collector in Li Metal Battery. *Angew. Chem. Int. Ed.* **63**, 2407064 (2024).
4. Kim, Y.-J. et al. Facet selectivity of Cu current collector for Li electrodeposition. *Energy Stor. Mater.* **19**, 154-162 (2019).
5. Kim, M.-H. et al. Horizontal lithium growth driven by surface dynamics on single crystal Cu(111) foil. *Energ. Environ. Sci.* **17**, 6521-6532 (2024).
6. Gu, Y. et al. Lithiophilic Faceted Cu(100) Surfaces: High Utilization of Host Surface and Cavities for Lithium Metal Anodes. *Angew. Chem. Int. Ed.* **58**, 3092-3096 (2019).
7. Ishikawa, K. et al. Crystal Orientation Dependence of Precipitate Structure of Electrodeposited Li Metal on Cu Current Collectors. *Cryst. Growth Des.* **17**, 2379-2385 (2017).